# Hippocampal and orbitofrontal neurons contribute to complementary aspects of associative structure

Huixin Lin [1,2] & Jingfeng Zhou [2] ✉

The ability to establish associations between environmental stimuli is fundamental for higher-order brain functions like state inference and generalization. Both the hippocampus and orbitofrontal cortex (OFC) play pivotal roles in this, demonstrating complex neural activity changes after associative learning. However, how precisely they contribute to representing learned associations remains unclear. Here, we train head-restrained mice to learn four 'odor-outcome' sequence pairs composed of several task variables—the past and current odor cues, sequence structure of 'cue-outcome' arrangement, and the expected outcome; and perform calcium imaging from these mice throughout learning. Sequence-splitting signals that distinguish between paired sequences are detected in both brain regions, reflecting associative memory formation. Critically, we uncover differential contents in represented associations by examining, in each area, how these task variables affect splitting signal generalization between sequence pairs. Specifically, the hippocampal splitting signals are influenced by the combination of past and current cues that define a particular sensory experience. In contrast, the OFC splitting signals are similar between sequence pairs that share the same sequence structure and expected outcome. These findings suggest that the hippocampus and OFC uniquely and complementarily organize the acquired associative structure.

Our brain constantly aims to form associations between environmental stimuli and organize them into mental constructs reflecting context, cognitive maps, and schemas—structures enabling higher-order brain functions like state inference and generalization[1–3]. Of particular importance in establishing such associations are the hippocampus and OFC, albeit in different domains. The hippocampus is recognized for its roles in spatial navigation and episodic memory[4,5], responding to sensory stimuli based on the specific time, location, and context[6–8] and facilitating rapid associations to form maps and memories related to specific experiences[9–11]. In contrast, the OFC has been primarily studied for its involvement in outcome representation[12–15] to support prediction and outcome-guided behaviors[16,17].

Although mostly from independent studies, similarities between the hippocampus and OFC have been increasingly suggested[18]. Notably, both regions have been proposed to hold a predictive map[19–22] and encode comprehensive information concerning past, present, and anticipated sensory stimuli influenced by temporal, spatial, and contextual factors[23–26]. Both are crucial for animals to identify and navigate partially observable task states within their environment[9,22,27,28]. Neural recordings show that both hippocampal and OFC neurons encode abstract values and schemas through generalization over similar experiences[15,29–32]. Given these observed similarities, it is essential to compare their neural activities directly within the same experimental settings to parse out their functional differences.

[1]Academy for Advanced Interdisciplinary Studies, Peking University, Beijing 100871, China. [2]Chinese Institute for Brain Research, Beijing 102206, China.
✉e-mail: jingfeng.zhou@cibr.ac.cn

Crucially, underlying these intricate brain functions are the fundamental roles of the hippocampus and OFC in representing basic associations, promoting us to dissect the precise nature of their neural activities related to learned associative structure—by investigating how they generalized across 'cue-outcome' associations under the influence of task variables including the past and current sensory cues, expected outcome, and the sequence structure defined as particular 'cue-outcome' arrangement. Specifically, we conducted calcium imaging in the hippocampus and OFC from separate head-restrained mice learning multiple 'odor-outcome' sequence pairs, within each pair sharing an overlapping epoch and across pairs affected by the above task variables. The emergence of neurons selective to paired sequences at the overlapping epoch—referred to as 'splitting signals', indicated associative memory formation. By examining how these splitting signals generalized between sequence pairs, we determined whether and to what extent the encoded associative structure incorporated information about the past and current cues, sequence structure, and expected outcome.

Our analyses revealed that the splitting signals in the hippocampus were prominently influenced by a blend of past and current cues, with a stronger impact from the current than the past cues; meanwhile, the splitting signals in the OFC were more generalizable between sequence pairs sharing the same sequence structure and expected outcome. This contrast suggests that these two regions play distinctive yet complementary roles in organizing learned associations, potentially working together to support more complex brain functions.

## Results

### Imaging CA1 and OFC neurons from mice learning association problems

Calcium imaging with gradient-index (GRIN) lenses was performed in the CA1 and OFC of separate head-restrained mice while they were learning two 'cue-outcome' association problems (Problem 1 and 2; Fig. 1a, b). Each problem employed a unique set of odor cues organized into four 'odor-outcome' sequences, namely a+, a−, b+, and b−. The + and − symbols indicate whether a drop of sucrose solution was delivered at the end of the sequence. Between the first odor cue and the outcome existed a second, overlapping odor cue for the sequence pair a+ vs. a−, but an overlapping delay for b+ vs. b−, such that this task design resulted in a sequence structure—meaning how the 'cue-outcome' was arranged—shared between sequence pairs 1a+ vs. 1a− and 2a+ vs. 2a− due to the overlapping odors, and another sequence structure shared between sequence pairs 1b+ vs. 1b− and 2b+ vs. 2b− due to the overlapping delay (i.e., sequence pair type a and b; Fig. 1b). Using this task design, we intended to search for neural signals distinguishing two sequences within each pair, particularly at the overlapping epoch, as an indication of associative learning. We then examined how such neural signals generalized between four sequence pairs (1a+ vs. 1a−, 1b+ vs. 1b−, 2a+ vs. 2a−, and 2b+ vs. 2b−) under the influences of the past (the first) and current (the second) cues, sequence structure, and expected outcome.

Over 12 days of learning Problem 1, mice ($n = 7$, CA1; $n = 4$, OFC) gradually developed differential anticipatory licking behaviors before reward delivery in response to reward and non-reward odor cues in both sequence pairs (1a+ vs. 1a− and 1b+ vs. 1b−) and reached an asymptote for each pair (brain region: $F_{(1,225)} = 0.61$, $p = 0.44$, sequence pair: $F_{(1,225)} = 26.33$, $p < 0.0001$, day: $F_{(11,225)} = 13.26$, $p < 0.0001$, three-way ANOVA; Fig. 1c, d and Supplementary Fig. 1). After training with Problem 2 for five days, these mice were subsequently placed on both problems for 12 days (brain region: $F_{(1,368)} = 32.22$, $p < 0.0001$, sequence pair: $F_{(3,368)} = 30.36$, $p < 0.0001$, day: $F_{(11,368)} = 1.77$, $p = 0.058$, three-way ANOVA; Fig. 1c, d and Supplementary Fig. 1).

Throughout training, we imaged neuronal calcium transients indicated by GCaMP6 fluorescent signals from these mice (CA1: 1612.7 ± 201 neurons per day from 7 mice, GCaMP6m expressed by AAV vectors, AAV-hSyn-GCaMP6m; OFC: 442.9 ± 55 neurons per day from 4 mice, GCaMP6f expressed in transgenic mice, Thy1-GCaMP6f; mean ± SD; Supplementary Fig. 2 and Supplementary Table 1); and carried out calcium signal deconvolution to estimate spike rates for further analysis (Fig. 1e, f).

### Emergence of sequence-splitting neurons reflects associative memory formation

With imaging data during the mice learning Problem 1, we focused on learning-related changes in differential neural activities to paired sequences (i.e., 1a+ vs. 1a−, 1b+ vs. 1b−, and 1a+ vs. 1b+) as an indication for the formation of associative memory, specifically at the sequence-overlapping epochs where the mice had to rely on an internal representation of sequence information but not merely external sensory stimuli, analogous to running through the common arm in a T-maze. Such sequence-selective neurons were defined as 'sequence-splitting neurons' or 'splitting neurons' for short.

After observing large fractions of task-activated neurons in both brain regions (CA1: 62.81% on Day 1 and 86.11% on Day 12, OFC: 72.85% on Day 1 and 86.01% on Day 12, *$p < 0.01$, Wilcoxon rank sum test; Supplementary Fig. 3) while mice were learning Problem 1, we started with a search for three types of sequence-splitting neurons: odor-splitting neurons that discriminated the sequence pair 1a+ vs. 1a− at 4–7 s, delay-splitting neurons that discriminated 1b+ vs. 1b− at 4–7 s, and reward-splitting neurons that discriminated 1a+ vs. 1b+ at 7–10 s (Fig. 2a), and we found the existence of all three types of splitting neurons in both CA1 and OFC (Fig. 2b, c and Supplementary Fig. 4). To determine whether anticipatory licking per se could fully account for the emergence of splitting neurons, we tested whether there were significant correlations between neural activities and the lick rate or lick latency. Our analyses showed that on average only a small subset of neurons exhibited such a correlation during learning (lick rate: 3.04 ± 0.39% in CA1 and 4.67 ± 0.57% in OFC, lick latency: 3.09 ± 0.42% in CA1 and 4.45 ± 0.59% in OFC, *$p < 0.01$, Pearson's correlation; Supplementary Fig. 5).

To characterize changes in splitting neurons during learning, we calculated the fractions of neurons selective to paired sequences (1a+ vs. 1a−, 1b+ vs. 1b−, and 1a+ vs. 1b+) at different time points during a trial (Fig. 3a, b). In both CA1 and OFC, we observed increased fractions of splitting neurons triggered at the second odor or delay epoch as mice gradually learned to discriminate paired sequences differing in reward availability (1a+ vs. 1a− and 1b+ vs. 1b−) despite the absence of sequence-discriminative information during these epochs. Subsequent quantitative analysis focused on the overlapping epoch (4–7 s) revealed a substantial increase in the fractions of odor- and delay-splitting neurons in both CA1 and OFC after learning, and the fractions of both types of splitting neurons were higher in OFC than CA1 (odor-splitting, brain region: $F_{(1, 97)} = 35.31$, $p < 0.0001$, day: $F_{(11, 97)} = 8.42$, $p < 0.0001$, interaction: $F_{(11, 97)} = 1.86$, $p = 0.054$; delay-splitting, brain region: $F_{(1, 97)} = 330.9$, $p < 0.0001$, day: $F_{(11, 97)} = 9.99$, $p < 0.0001$, interaction: $F_{(11, 97)} = 2.78$, $p = 0.004$, two-way ANOVA; Fig. 3c, d). However, there was no significant learning-related changes in the fractions of reward-splitting neurons during the outcome epoch (7–10 s) for paired sequences 1a+ vs. 1b+ (Fig. 3a–d), and slightly more reward-splitting neurons in CA1 than OFC (brain region: $F_{(1, 97)} = 13.46$, $p = 0.0004$, day: $F_{(11, 97)} = 0.85$, $p = 0.594$, interaction: $F_{(11, 97)} = 0.99$, $p = 0.464$, two-way ANOVA; Fig. 3c, d).

Next, we used a discrimination index (DI) to quantify each neuron's strength of selectivity to the paired sequences at overlapping epochs (odor or delay: 4–7 s, reward: 7–10 s; Fig. 3e–h). A higher DI reflects a greater discrimination of paired sequences, indicating

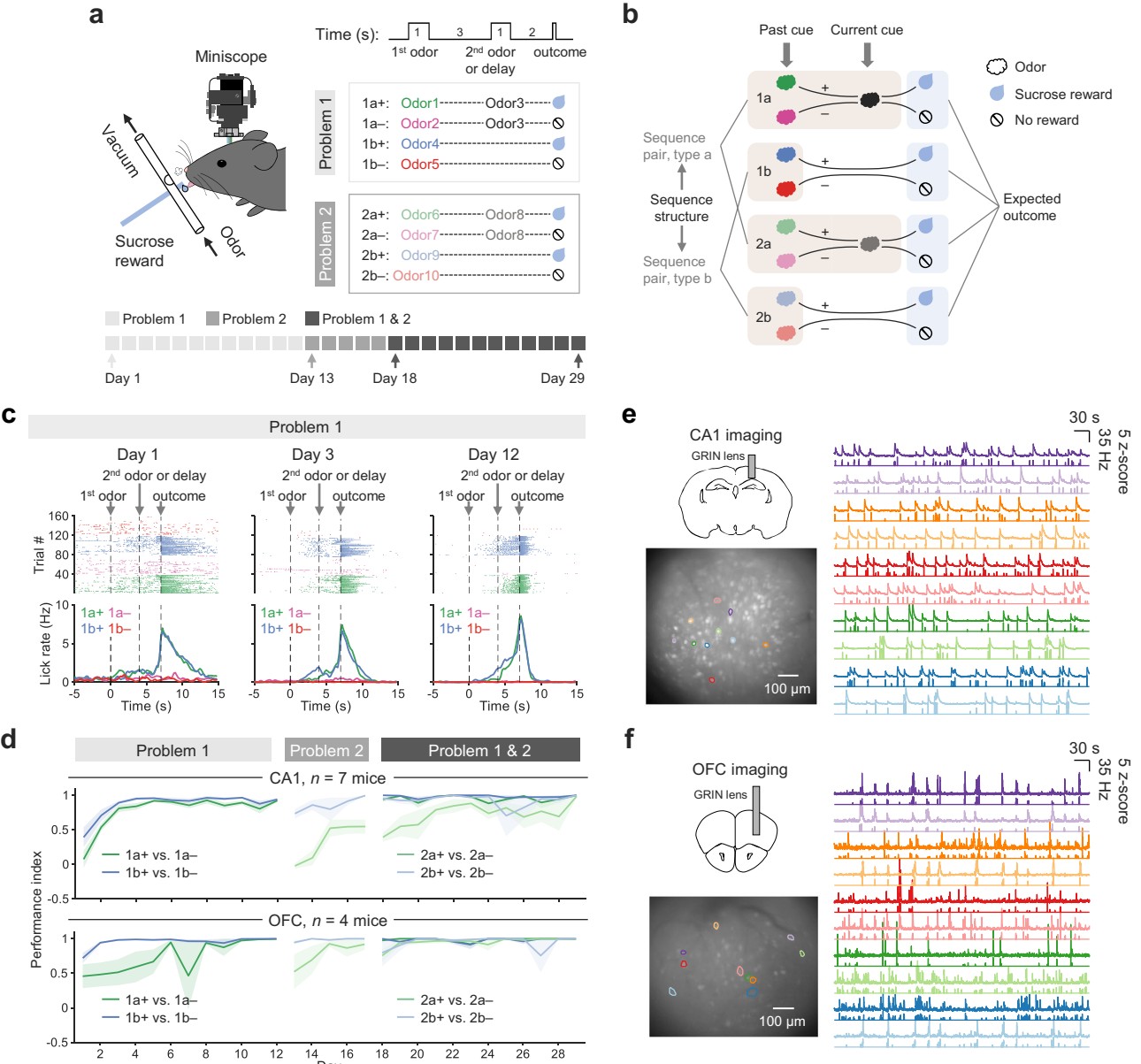

**Fig. 1 | Task design, behavior, and calcium imaging. a** Calcium imaging was performed on head-restrained mice learning two odor-outcome association problems, consisting of four paired sequences (1a+ vs. 1a−, 1b+ vs. 1b−, 2a+ vs. 2a−, and 2b+ vs. 2b−). **b** Task variables reflecting specific aspects of learned associative structure−including the past and current cues, expected outcome, and sequence structure that was shared between sequence pairs of the same type (a or b: with or without a common odor, respectively). **c** Behavioral example from one mouse learning Problem 1. Rasters indicate licks. The dashed lines indicate the onsets of the first odor, second odor, and outcome. Each color represents one trial type. **d** Performance index = [LR(+) − LR(−)]/[LR(+) + LR(−)]. LR indicates anticipatory lick rate (with 0 s being the onset of the first odor); (+) and (−) mean reward and no-reward, respectively. Data are presented as mean ± SEM. **e** Calcium imaging of dorsal CA1 in mice infused locally with AAV viruses expressing GCaMP6m. The projection of the maximal intensity from one example session (Mouse #1, Day 1) out of 154 CA1-imaging sessions highlights ten identified neurons represented in different colors; traces show their fluorescence signals (upper line of each color) and deconvoluted spike rates (lower line of each color) of consecutive 500 s in the example session. **f** Calcium imaging of OFC in Thy1-GCaMP6f mice, with the example session from Mouse #10 on Day 1 out of 114 OFC-imaging sessions. Source data are provided as a Source Data file.

stronger splitting signals. A larger number of neurons exhibited high DIs, thus demonstrating stronger odor-splitting signals in both CA1 and OFC (Fig. 3e–h, left). Similar trends were observed for the delay-splitting signals in both brain regions, although they did not reach statistical significance (Fig. 3e–h, middle). In contrast, these neurons displayed decreased splitting signals in response to paired sequences leading to the same reward (1a+ vs. 1b + ; Fig. 3e–h, right), indicating neural generalization between sequences sharing the same outcome. Furthermore, the heightened odor- and delay-splitting signals in both CA1 and OFC were not solely attributed to the increase in the fraction

of splitting neurons but were also partially explained by their heightened splitting strength (Supplementary Fig. 6).

In addition, we tracked the same cell populations throughout mice learning Problem 1 across days to characterize the consistency of odor- and delay-splitting neurons (Supplementary Fig. 7). Overall, while the splitting neurons in both brain regions were dynamically changing, manifested as 'adding' and 'dropping', these neurons were relatively less stable in the CA1 than in the OFC (overlapping fractions with one-day gap: CA1, 19.94%, OFC, 46.14%, odor- and delay-splitting averaged; Supplementary Fig. 8a–c), and in both brain regions, the

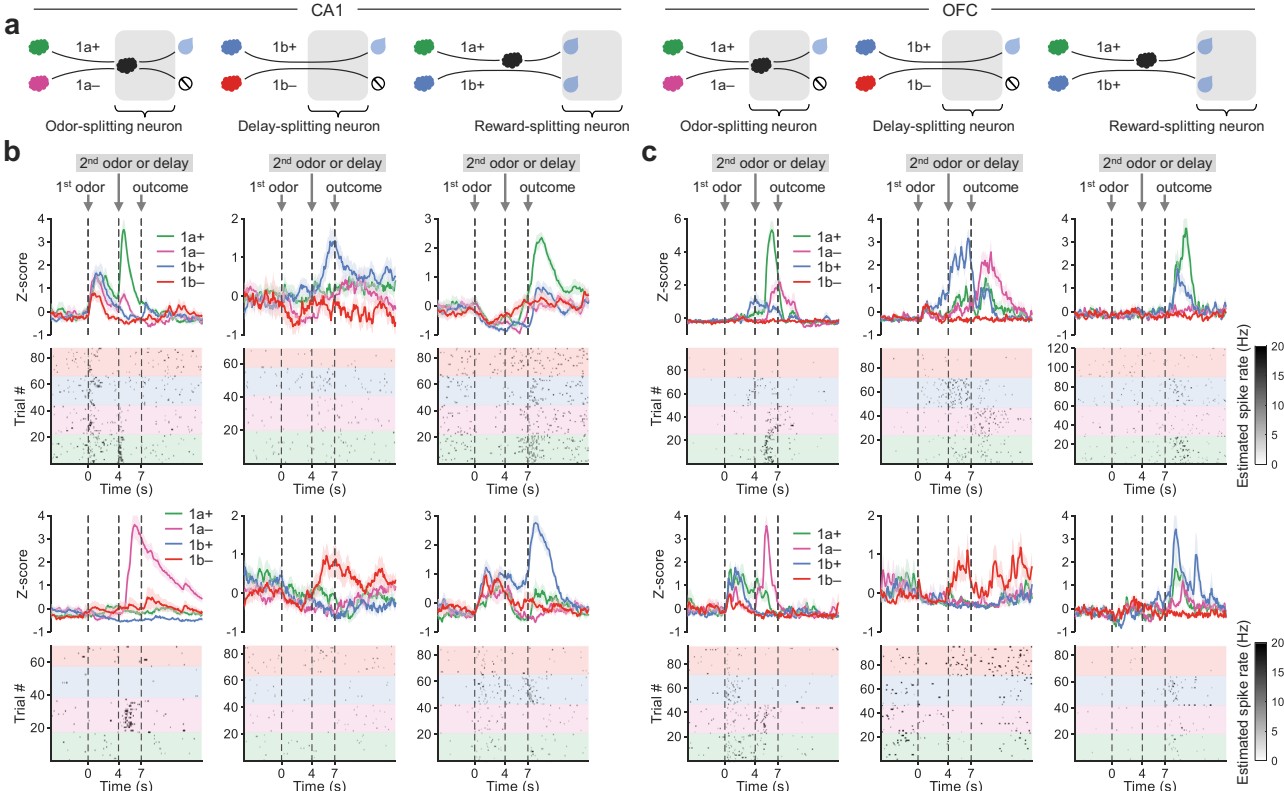

**Fig. 2 | Example sequence-splitting neurons in CA1 and OFC. a** Schematics illustrate the definitions of three types of splitting neurons. Neurons responding differentially to sequences a+ vs. a− at 4−7 s were identified as odor-splitting neurons, to b+ vs. b− at 4−7 s as delay-splitting neurons, and to a+ vs. b+ at 7−10 s as reward-splitting neurons (*$p$ < 0.01, Wilcoxon rank sum test). **b**, **c** Example splitting neurons in CA1 (**b**) and OFC (**c**) showing differential activities to paired sequences. Data are presented as mean ± SEM. Source data are provided as a Source Data file.

splitting signals became more stable during the late phase of learning Problem 1 (overlapping fractions between Day 1 and 2: CA1, 7.57%, OFC, 13.15%; between Day 11 and 12: CA1, 22.29%, OFC, 52.77%, odor- and delay-splitting averaged; Supplementary Fig. 8d, e). Besides, different types of splitting neurons in both regions were interchangeable across days (odor- to delay-splitting conversion: CA1, 23.29%, OFC, 50.16%; delay- to odor-splitting conversion: CA1, 24.55%, OFC, 27.84%, averaged across days; Supplementary Fig. 8f, g). Similar results were observed when mice were performing Problem 1 and 2 together (Supplementary Fig. 9−11).

Although there were differences in the exact fractions of splitting neurons and their stabilities across days, these results highlighted that overall, the CA1 and OFC exhibited similar emergence of sequence-splitting activities during mice's learning of paired sequences, indicating their shared involvement in forming associative memory.

### Differential generalization of splitting signals in CA1 and OFC of mice learning Problem 1

After observing similar emergence of splitting neurons in the hippocampus and OFC as mice learning Problem 1, we sought to compare their splitting signals at the neural population level, focusing on odor and delay epochs (4−7 s; Fig. 4a, b). We pooled imaged neurons across mice for each brain region and each day as pseudo-ensembles; and then randomly selected subsets of neurons or used all neurons daily to train support vector machine (SVM) decoders and tested how accurately these decoders could correctly decode paired sequences (1a+ vs. 1a− and 1b+ vs. 1b−) with leave-one-out cross-validation (shown are decoding on Day 1, 3, and 12; Fig. 4a, b); higher accuracy indicates stronger neural population-level splitting signals. With a lower neuron number (e.g., 80 neurons), the decoding accuracy for both sequence pairs (1a+ vs. 1a− and 1b+ vs. 1b−) gradually increased over training

days, while with all imaged neurons used in the training of decoders, the decoding accuracy quickly reached the asymptote (100%) around Day 3 (odor-splitting, Fig. 4a; delay-splitting, Fig. 4b), demonstrating robust splitting signals at the neural population level throughout learning.

Training an SVM decoder in the neural activity space returned a hyperplane that best separated trials belonging to paired sequences (1a+ vs. 1a− or 1b+ vs. 1b−; Fig. 4c). Perpendicular to the decoder hyperplane, the associated normal vector−namely the splitting vector−represented a neural signal separating paired sequences in the population neural activity space. More specifically, the odor-splitting vector was the normal vector to separate paired sequences with the presence of the second odor cue (a+ vs. a−), and the delay-splitting vector was the normal vector to separate paired sequences with a long delay (b+ vs. b−). Finally, to quantify the coding similarity between sequence pairs (odor-splitting: 1a+ vs. 1a−, delay-splitting: 1b+ vs. 1b−), we calculated the cosine of the angle between the two splitting vectors (i.e., cosine similarity from −1 to 1, a higher number indicates higher coding similarity); and observed a low similarity between odor- and delay-splitting signals in the CA1, but a high similarity in the OFC (brain region: $F_{(1, 7176)}$ = 94356.79, $p$ < 0.0001, day: $F_{(11, 7176)}$ = 5485.35, $p$ < 0.0001, interaction: $F_{(11, 7176)}$ = 1122.46, $p$ < 0.0001, two-way ANOVA; Fig. 4c), which was consistent with an analysis using the multidimensional scaling (MDS; Supplementary Fig. 12). In addition, we replicated these coding similarity analyses with neural ensembles simultaneously imagined from individual sessions (brain region: $F_{(1, 97)}$ = 78.8, $p$ < 0.0001, day: $F_{(11, 97)}$ = 7.65, $p$ < 0.0001, interaction: $F_{(11, 97)}$ = 1.7, $p$ = 0.084, two-way ANOVA; Supplementary Fig. 13).

These results suggested that the sequence-splitting signals in the OFC were more likely than the hippocampus to generalize from one sequence pair to another, which promoted us to analyze imaging data

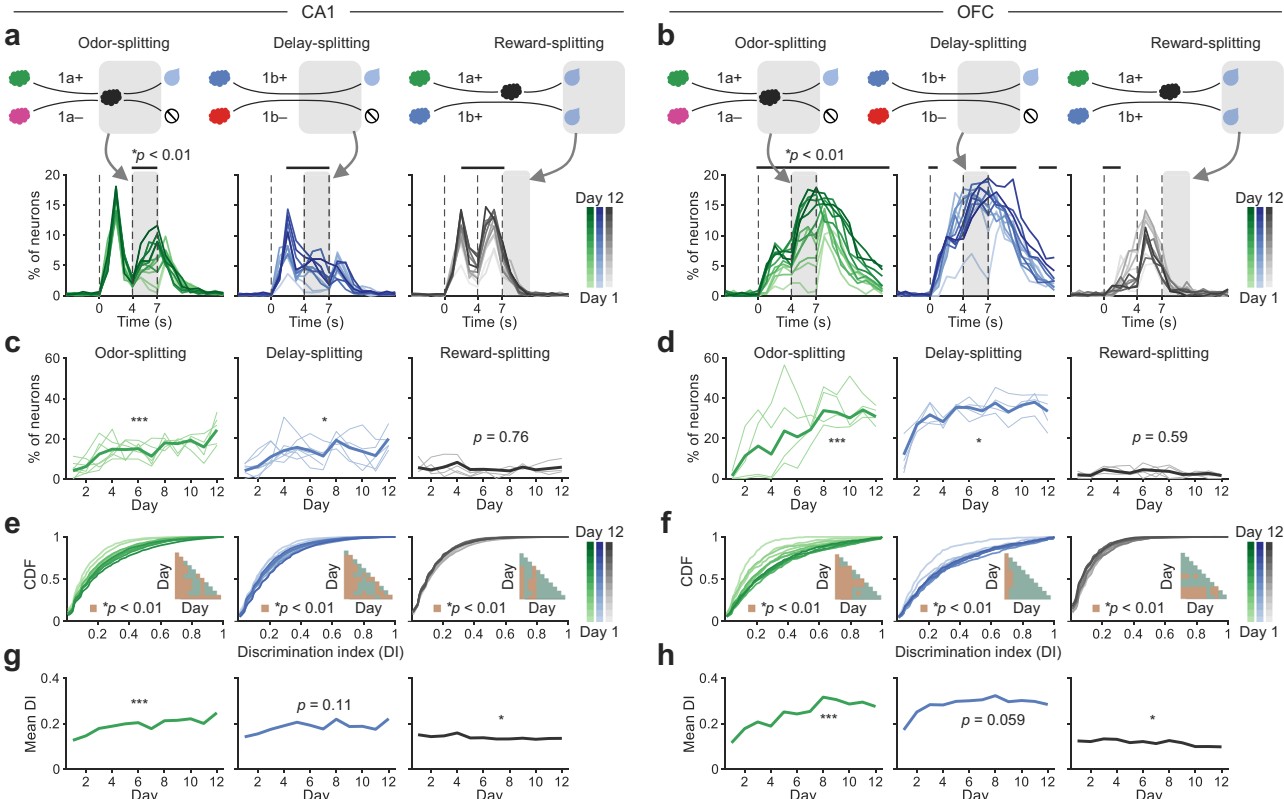

**Fig. 3 | Learning-related changes of splitting neurons in Problem 1. a** Percentage of neurons showing differential activities to paired sequences in CA1 (the denominator was all neurons imaged on a given day, same as following unless otherwise specified; bin size: 1 s; *$p < 0.01$, Wilcoxon rank sum test) throughout learning. Gray-shaded areas highlight the overlapping epoch in each sequence pair. The correlations between the percentages of splitting neurons and training days were tested by Spearman's rank correlation. **b** OFC neurons with the same analysis as in (**a**). **c** The odor-, delay- and reward-splitting neuron fractions during learning in CA1 (*$p < 0.05$, ***$p < 0.001$, Spearman's rank correlation). Bold lines indicate

splitting neurons imaged from all mice, and thin lines indicate those from individual mice. **d** OFC neurons with the same analysis as in (**b**). **e** The cumulative distribution function (CDF) of the discrimination index (DI) of CA1 neurons at overlapping odor, delay, and reward periods. The triangles show the pairwise comparison between the DI distributions of two training days (*$p < 0.01$, Kolmogorov-Smirnov test). **f** OFC neurons with the same analysis as in (**e**). **g** The mean DI of all CA1 neurons during learning (*$p < 0.05$, ***$p < 0.001$, Spearman's rank correlation). **h** OFC neurons with the same analysis as in (**g**). All statistical tests used were two-sided. Source data are provided as a Source Data file.

collected from mice performing both problems with all task variables, including past and current cues, sequence structure, and expected outcome. As aforementioned, by testing the generalization between splitting signals, we would be able to determine to what extent these task variables contribute to the composition of the learned associative structure in each brain region.

## Model simulation demonstrates task variables distinctly affect splitting signal generalization

Before testing with the actual data, we built a computational model based on the task design to understand how exactly different task variables would affect splitting generalization. Specifically, this model included four task variables: past cue, current cue, sequence structure, and expected outcome (Fig. 5a). For each of four paired sequences ($i = 1$: 1a+ vs. 1a−, $i = 2$: 2a+ vs. 2a−, $i = 3$: 1b+ vs. b−, $i = 4$: 2b+ vs. 2b−), we used a set of zero-mean orthonormal vectors ($\mathbf{a}_i$, $\mathbf{b}_i$, $\mathbf{t}_i$, $\mathbf{o}_i$) to represent the impact of four task variables on the sequence-splitting signals; each vector contained 100 elements representing 100 neurons that formed a high-dimensional neural activity space. The vectors $\mathbf{a}_1$, $\mathbf{a}_2$, $\mathbf{a}_3$, $\mathbf{a}_4$, $\mathbf{b}_1$, and $\mathbf{b}_2$ were uncorrelated ($r = 0$), corresponding to past ($\mathbf{a}_1$, $\mathbf{a}_2$, $\mathbf{a}_3$, $\mathbf{a}_4$) and current ($\mathbf{b}_1$, $\mathbf{b}_2$) odor cues. $\mathbf{b}_3$ and $\mathbf{b}_4$ were zero vectors, indicating mere delay time in their corresponding paired sequences. $\mathbf{t}_1$ and $\mathbf{t}_2$ were correlated ($r = 1$); $\mathbf{t}_3$ and $\mathbf{t}_4$ were correlated ($r = 1$) because of their respective common sequence structures. The vectors $\mathbf{o}_1$, $\mathbf{o}_2$, $\mathbf{o}_3$, $\mathbf{o}_4$ were correlated ($r = 1$) because of the same expected outcome.

Each splitting vector, $\mathbf{s}_i$ ($i = 1, 2, 3, 4$), unit-length, was a weighted linear summation of four basis vectors ($\mathbf{a}_i$, $\mathbf{b}_i$, $\mathbf{t}_i$, $\mathbf{o}_i$), with α, β, γ, δ being their corresponding weights—reflecting how the four task variables collectively influenced each splitting vector, determining both its direction and magnitude. Thus, $\mathbf{s}_i = \alpha \cdot \mathbf{a}_i + \beta \cdot \mathbf{b}_i + \gamma \cdot \mathbf{t}_i + \delta \cdot \mathbf{o}_i$, $i = 1, 2, 3, 4$. With this model, we could then predict the overlapping fractions of splitting neurons (Fig. 5b, c) and coding similarity (Fig. 5d) between splitting vectors or signals ($\mathbf{s}_1$, $\mathbf{s}_2$, $\mathbf{s}_3$, $\mathbf{s}_4$) by a given set of weight parameters. The simulation results showed that a high α (0.98; β, γ, δ = 0.11) resulted in dissimilarities between the four splitting signals; a high β (0.98; α, γ, δ = 0.11) resulted in a dissimilarity between odor-splitting signals ($\mathbf{s}_1$ and $\mathbf{s}_2$), but a similarity between delay-splitting signals ($\mathbf{s}_3$ and $\mathbf{s}_4$); a high γ (0.98; α, β, δ = 0.11) resulted in similarities between splitting signals sharing the same type of sequence structure (type a: $\mathbf{s}_1$ and $\mathbf{s}_2$; type b: $\mathbf{s}_3$ and $\mathbf{s}_4$); a high δ (0.98; α, β, γ = 0.11) resulted in similarities between the four splitting signals. Thus, the model simulation demonstrated precisely how these task variables would differentially affect splitting signal generalization.

## Influence of task variables on splitting signal generalization reveals that CA1 and OFC emphasize different aspects of associative structure

To examine whether the simulated coding similarity patterns did exist in CA1 and OFC neuronal activities, we measured the overlapping fractions of odor- and delay-splitting neurons between two problems

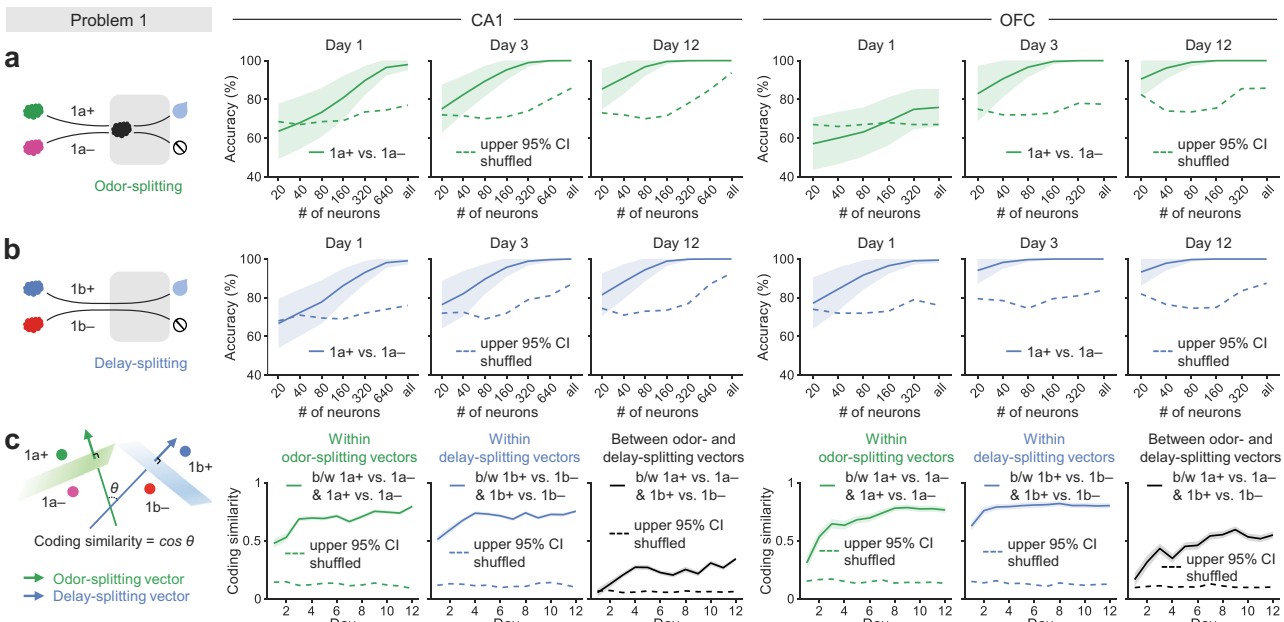

**Fig. 4 | Splitting signals at the level of neural ensembles in Problem 1. a, b** SVM decoding of paired sequences with varying neuron numbers. The decoders were trained and tested with neural activities sampled during odor-splitting (**a**) or delay-splitting (**b**) period (4–7 s). Shaded areas indicate mean ± SD of decoding accuracy from 500 repeats. Dashed lines indicate the upper bound of the 95% confidence interval (CI) of the accuracy resulted from decoding on label-shuffled data. **c** The coding similarity is the cosine of the angle between two splitting vectors perpendicular to the SVM hyperplanes that separate sequences at overlapping epochs. b/w: between. Shaded areas are SD from 300 repeats. Dashed lines show the upper bound of the 95% CI with label-shuffled data. Source data are provided as a Source Data file.

as well as coding similarities between the four splitting signals for paired sequences ($s_1$: 1a+ vs. 1a−, $s_2$: 2a+ vs. 2a−, $s_3$: 1b+ vs. 1b−, and $s_4$: 2b+ vs. 2b−; Fig. 6).

In the CA1, although the overlapping fraction of odor-splitting neurons slightly increased during learning, only a small proportion of odor-splitting neurons were shared between the two problems by the last day of training (~10.2%; denominator being all odor-splitting neurons in both problems). In comparison, there were more delay-splitting neurons shared between the two problems during learning (~24.0%; Fig. 6a and Supplementary Fig. 14a, b). This observation matched the model simulation with a high β, suggesting the current overlapping odor cue may provide a unique temporally proximal sensory context that reduced odor-splitting signal generalization in the CA1. To further confirm this at the neural ensemble level, we computed the coding similarity between splitting signals under the verified prerequisite that sequences within each sequence pair were decodable throughout learning (Fig. 6b). And indeed, we observed a higher between-problem similarity of delay-splitting signals ($s_3$ and $s_4$) than that of the odor-splitting signals ($s_1$ and $s_2$; Fig. 6c and Supplementary Fig. 15a).

By contrast, in the OFC, we saw quite different patterns of splitting similarities. First, the two problems shared many odor-splitting neurons (33.0–59.8%) as well as delay-splitting neurons (47.9–64.9%) throughout learning (Fig. 6d and Supplementary Fig. 14c, d). Second, under the condition of nearly perfect decoding of paired sequences with daily imaged OFC neurons (Fig. 6e), the four splitting signals were generally similar across all sequence pairs, with higher similarities between sequence pairs that shared the same type of sequence structure (type a, odor-splitting: $s_1$ and $s_2$; type b, delay-splitting: $s_3$ and $s_4$) than otherwise (Fig. 6f and Supplementary Fig. 15b). These findings were consistent with the model simulation with a high γ and δ, suggesting that the sequence structure and expected outcome, other than the past and current cues (characterized by α and β), predominantly formed the content of splitting signals in the OFC. In addition, for both CA1 and OFC, respective

similar results were observed with the coding similarity analyses using neural ensembles imaged from individual sessions (Supplementary Fig. 16).

Finally, when fitting the model with the imaging data, we obtained almost identical coding similarity patterns as with the data for both CA1 and OFC (Fig. 7a–c). Through comparing the recovered parameters (α, β, γ, δ), we found that effects from all the task variables on splitting signals were evident in both brain regions (Fig. 7d, e and Supplementary Fig. 17). However, the past (α) and current (β) cues exerted much stronger modulation on the splitting signals in CA1 than OFC (α, brain region: $F(1, 2387) = 1207.28$, $p < 0.0001$; β, brain region: $F(1, 2387) = 9829.37$, $p < 0.0001$; two-way ANOVA with 'brain region' and 'day'), with the current cue more prominent than the past cue (α vs. β: $F(1, 2387) = 3587.31$, $p < 0.0001$; two-way ANOVA with 'α vs. β' and 'day'), meaning that CA1 splitting signals reflected mixed influence from both the current cue and decaying past cue, consistent with a proposed function of the hippocampus in signaling a temporally convoluted sensory experience, or namely temporal context[9]. In contrast, the sequence structure (γ) and outcome prediction (δ) affected the splitting signals in the OFC more strongly than in the CA1 (γ, brain region: $F(1, 2387) = 17208.06$, $p < 0.0001$; δ, brain region: $F(1, 2387) = 46940.6$, $p < 0.0001$; two-way ANOVA with 'brain region' and 'day'), suggesting the potentially more important role of the OFC in state generalization.

Note that although we used two different GCaMP6 variants (GCaMP6m and GCaMP6f), deconvolution of calcium signals was helpful to mitigate their calcium-unbinding kinetic differences as we found no significant differences in estimated spike rates between the two indicators in CA1 (Supplementary Fig. 18a–c), and critically, we reproduced the described CA1 imaging results with a new cohort of Thy1-GCaMP6f mice ($n = 4$; Supplementary Fig. 18d–n), demonstrating the robustness of these findings, regardless of the specific GCaMP6 variants used.

In sum, through a combination of behavioral task design, calcium imaging, neural data analyses, and computational modeling,

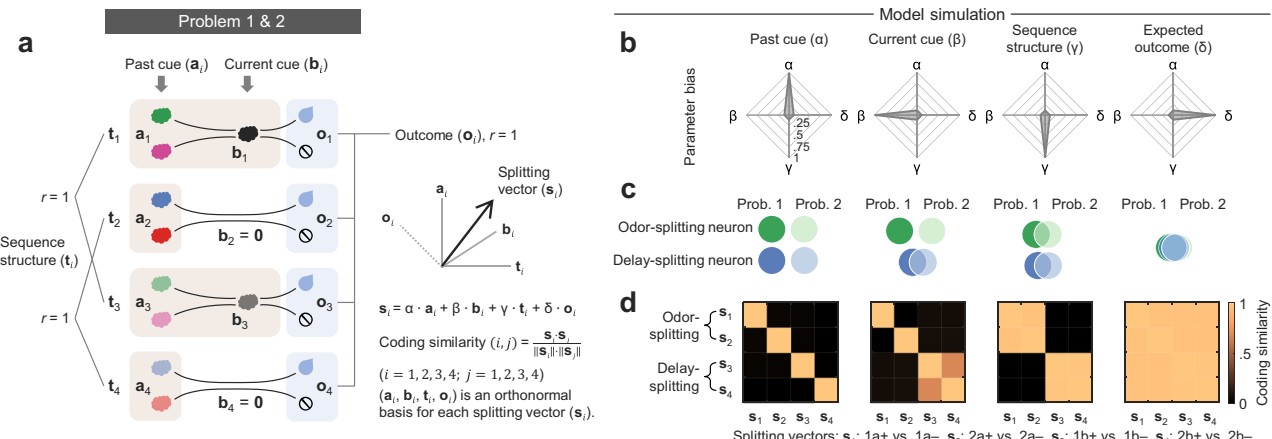

**Fig. 5 | Model simulation on how task variables affect generalization between splitting signals. a** Decomposing the splitting signal ($\mathbf{s}_i$) for each sequence pair into four task variables: past cue ($\mathbf{a}_i$), current cue ($\mathbf{b}_i$), sequence structure ($\mathbf{t}_i$), and expected outcome ($\mathbf{o}_i$). $i = 1$: 1a+ vs. 1a−, $i = 2$: 2a+ vs. 2a−, $i = 3$: 1b+ vs. 1b−, $i = 4$: 2b+ vs. 2b−. **a, b, t,** and **o** are 100-dimensional orthonormal vectors. $\mathbf{a}_1$ to $\mathbf{a}_4$ and $\mathbf{b}_1$ to $\mathbf{b}_2$ were uncorrelated ($r = 0$), and $\mathbf{b}_3$ to $\mathbf{b}_4$ were zero vectors. $\mathbf{t}_1$ and $\mathbf{t}_2$, $\mathbf{t}_3$ and $\mathbf{t}_4$, $\mathbf{o}_1$ to $\mathbf{o}_4$ were correlated ($r = 1$), respectively. Splitting signals (**s**) were formalized as a linear

combination of **a, b, t,** and **o** with α, β, γ and δ being their corresponding weights. **b** Radar plots indicate four types of weight biases toward α, β, γ, or δ, each of which has a range between 0 and 1. **c** Venn diagrams of overlapping splitting neurons predicted with parameter biases in (**b**). **d** Simulation of the coding similarity using four sets of parameters in (**b**). The coding similarity between two splitting signals was computed as cosine similarity. Source data are provided as a Source Data file.

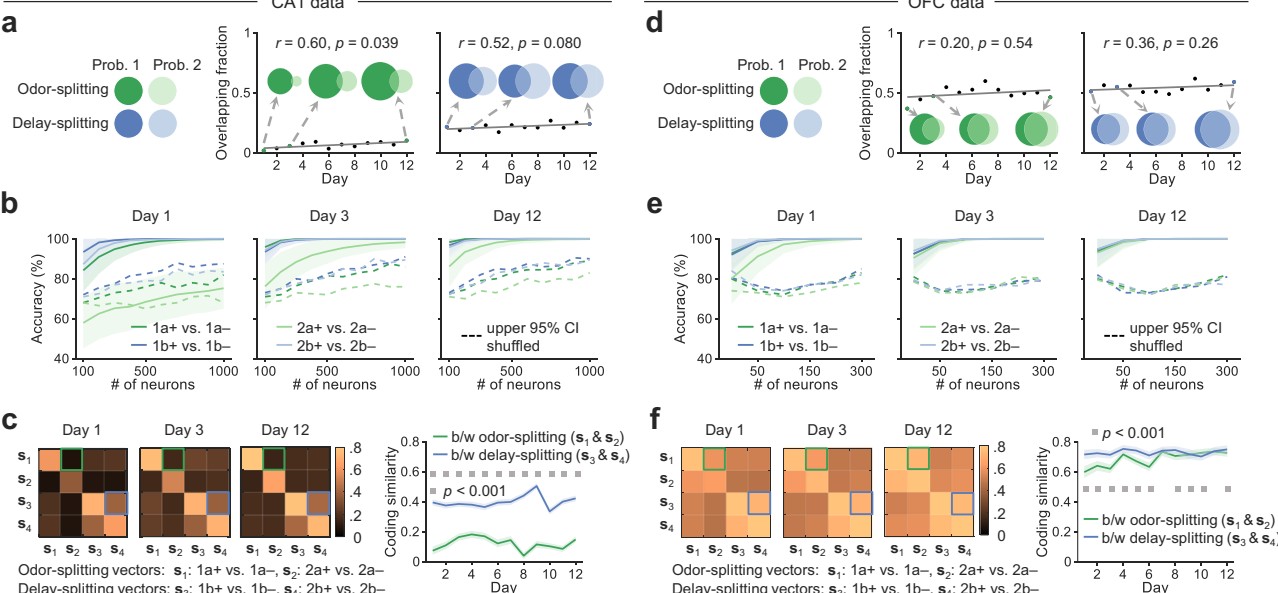

**Fig. 6 | Comparison of splitting signal generalization in CA1 and OFC neural activities. a** Overlapping fraction of splitting neurons in CA1 during learning. Venn diagrams show example sessions on Day 1, 3, and 12. Black lines show correlations between overlapping fractions and training days (Spearman's rank correlation). **b** SVM decoding for paired sequences during the sequence-overlapping epoch (4–7 s) with varied numbers of neurons sampled in CA1. Shaded areas mean the mean ± SD of decoding accuracy from 500 repeats. Dashed lines are the upper

bound of the 95% CI of the decoding accuracy performed on label-shuffled data. **c** Coding similarities of four splitting signals in CA1 during learning. b/w: between. Shaded areas mean the mean ± SD from 300 repeats. The significance of differences was examined by the permutation test (***$p < 0.001$, $10^5$ times). **d–f** OFC neurons with the same analyses as in **a–c**. All statistical tests used were two-sided. Source data are provided as a Source Data file.

we found that sequence-splitting signals emerged in neural activities of both the hippocampus and OFC to encode distinct yet complementary aspects of the learned associative structure, with the hippocampus emphasizing specific sensory experiences leading to future outcomes and OFC the sequence structure and expected outcome in a more abstract format. These results suggest specific roles of these two critical brain regions in organizing past experiences to support complex behaviors requiring flexible state discrimination and generalization.

## Discussion

In this study, by performing calcium imaging on mice learning different 'odor-outcome' sequence pairs, we compared the abilities of sequence-splitting signals in the hippocampus and OFC to generalize between these sequence pairs. While such splitting signals—reflecting learned associations—similarly emerged in both regions, they were different in how they were influenced by the past and current odor cues, sequence structure, and expected outcome. The hippocampal splitting signals were tied to temporally more proximal olfactory cues,

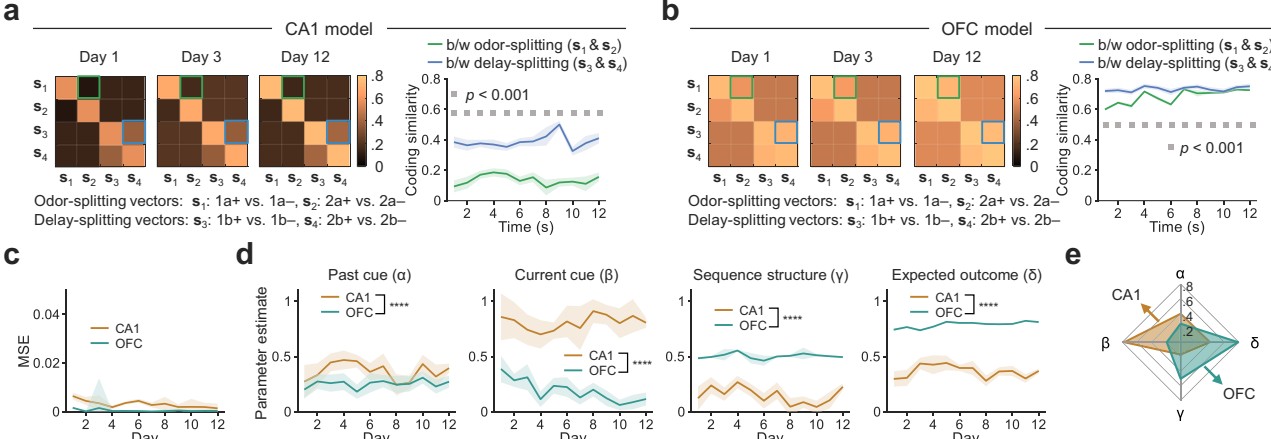

**Fig. 7 | Model fitting reveals CA1 and OFC emphasize different aspects of learned associations. a–b** Model-predicted coding similarities of four splitting signals in CA1 (**a**) and OFC (**b**) during learning. b/w: between. The annotations and statistics are the same as in Fig. 6c, f. **c** Mean squared error (MSE) between the predicted and actual coding similarities. Shaded areas show the mean ± SD from 100 repeats. **d** Recovered parameters to fit the model on each day. Shaded areas indicate the mean ± SD from 100 repeats. The significance of differences for each parameter was examined by two-way ANOVA ('brain region' and 'day', ****$p < 0.0001$). **e** Radar plot of parameter estimates averaged across days for CA1 and OFC. All statistical tests used were two-sided. Source data are provided as a Source Data file.

suggesting that the hippocampal associative structure incorporated recent past and current sensory experiences, in line with the role of hippocampus in signaling temporal context[4,9,11,33]. In contrast, the orbitofrontal splitting signals were similar between sequence pairs with the same sequence structure and expected outcome, implying more likely roles of the OFC in generalization and schema representations[15,34]. Thus, these results revealed differential contributions of the two brain regions to establishing associative structures.

Extensive research has been conducted on the hippocampus concerning spatial navigation, with the hippocampus believed to map the external context to a mental representation[1,35], and such cognitive mapping not merely mirrors the surrounding environment but involves the intricate reorganization of experiences into structured memories that reflect animals' experienced and inferred knowledge about the environment[3,36,37]. One prominent example is the hippocampal splitter cells in the T-maze and other similar mazes—these cells fire differently to overlapping trajectories differing in the animal's starting point, destination, or inference about the environment[9,38–41]. The current study observed non-spatial sequence-splitting cells in the hippocampus and the OFC, much like the hippocampal spatial splitter cells, which is consistent with some other findings. For instance, the human hippocampus can disambiguate overlapping episodic sequences[42]; the single-unit activities in the rat hippocampus represent the temporal order of odors and discriminate between overlapping odor sequences[26,43–46]. Our finding of non-spatial splitting signals could thus help further reconcile hippocampal functions in episodic memory and spatial navigation[4,37]. On the other hand, such splitting signals can be used as a neural proxy for associative memory with minimal sensory confounds[9]. Specifically, unlike place cells or sensory stimulus-responsive cells, these splitting signals hold an internal representation of sequence information latent and unobservable from current external sensory inputs, thus enabling us to extract specific as well as more abstract contents in learned associations.

Traditionally, OFC has been studied independently of the hippocampus, with the OFC mostly being focused on its role in outcome prediction within the non-spatial domain[14]. However, separate lines of research have recently converged and resulted in surprisingly similar ideas about the functions of the two brain regions[18,22,31,47,48]. Many loss-of-function studies have emphasized the necessity of the OFC in flexible and goal-directed behaviors where disambiguating different states, rules, or contexts is critical for behavioral performance[49–54].

Electrophysiological recordings in the OFC have also revealed encoding of comprehensive task information, including past sensory stimuli, predictive coding for future outcomes, latent stimulus-stimulus associations, temporal order, and non-spatial splitter cells that distinguish between overlapping events[15,55–58], which are similar to findings in the hippocampal studies[18,45]. Thus, a direct comparison between the two brain regions under the same experimental condition is desirable but rare in the literature[26]. By using the same task that involved sequence-splitting signals in both regions, the current study characterized their roles in representing different aspects of associative structure, highlighting the importance of using the same task to dissociate neural substrates for task and cognitive variables between different brain structures.

Our data showed that the past and current cues modulated the splitting signals in the hippocampus; this is consistent with previous findings showing that some task variables (e.g., reward value and accumulated evidence) encoded by the hippocampus nonlinearly interact with spatial information (i.e., conjunctive coding), which is thought to facilitate the formation of a particular spatial map[7,32,59]. Beyond these findings, our current finding is particularly intriguing because the splitting signals detected in this study represent a more abstract, non-spatial task variable—commonly recognized as a top-down predictive signal—beyond the perceptual and spatial domains. We report that such a presumed 'predictive signal' was strongly bound to the combination of past and current cues signaling a particular sensory experience. This finding suggests an extended hippocampal function, where beyond the conjunctive coding between sensory stimuli and space, the hippocampus might provide general computing machinery that blends various, both bottom-up and top-down, signals to generate unique, combinatorial, or intermingled representations that could be useful for context disambiguation and hidden-state inference[9].

However, such binding of splitting signals to recent sensory stimuli did not explain the splitting signals in the OFC as much as in the hippocampus. Instead, OFC splitting signals were shared between sequence pairs that were similar in sequence structure and expected outcome. While this is not surprising given that it is known that OFC represents outcome expectancy[14], it is, however, important to note that the OFC splitting neurons were more prominently shared between sequence pairs with the same sequence structure, even though all sequence pairs used different odor cues, which is consistent with our

recent finding that the rat OFC ensembles represent a task schema across learning problems[15]. Moreover, more recent work has discovered that the schema representation seems independent of a hippocampal output in well-trained rats or could even be boosted with a hippocampal output impairment during learning[34], suggesting that the information processing in the OFC and hippocampus could work independently to represent different aspects of the associative structure.

In this study, we used calcium imaging to proxy neural activities. While calcium imaging is a powerful tool useful in the current and other studies that aim to understand brain functions, it does have limitations, such as its indirect measure of neural activity, limited temporal resolution, nonlinear relationship between calcium transients and action potentials, and the use of GRIN lenses causing more damage to brain tissues than electrophysiological recordings. For instance, recent studies have directly compared the calcium and electrophysiological signals and found differences in their activities encoding task variables[60,61]. Another limitation of our current study is the lack of causal evidence to show that the different neural signals observed in the two brain regions are distinctively relevant to behaviors, which might require further modifications in the task design since the mice's anticipatory licking behaviors were not mandatory in the current Pavlovian setting but only reflective of their appreciation of the associative structure. Therefore, future studies should consider these limitations and confirm these findings with other techniques, including functional perturbation to gain a more comprehensive understanding of the currently reported results.

In sum, we observed sequence-splitting signals emerged similarly in the hippocampus and OFC during learning; however, such splitting signals in the two regions were different in their ability to generalize across sequence pairs, from which we discovered the activities in the hippocampus and OFC emphasize distinct components in the learned associative structure. These findings extend our understanding of how the two brain structures, critical for associative learning and memory, might differently and collaboratively organize associative structure. Further investigations exploring the temporally dynamic interdependence of neural representations for these complementary contents should help identify neural mechanisms regarding the interaction between different memory systems to support associative memory acquisition, consolidation, retrieval, and transfer[2].

## Methods

### Animal subjects
Seven male C57BL/6 J mice, six male and two female Thy1-GCaMP6f mice (C57BL/6J-Tg(Thy1-GCaMP6f)GP5.17Dkim/J, The Jackson Laboratory, 025393), aged 7–8 weeks at the start of the experiment, were used for behavioral training and calcium imaging. The mice were group-housed, up to 5 animals in one cage, on a 12-hour light and 12-hour dark cycle. Experiments were conducted during the dark cycle except for two mice for the CA1 imaging. The ambient temperature was 23–25 °C and the humidity was 40–50%. All animals were given free access to food and water until 2–3 days before behavioral training when they were water-restricted and maintained at least 80% of their free-drinking body weight. All behavioral experiments were carried out at CIBR. The Animal Care and Use Committee (ACUC) at CIBR approved the animal care and experimental procedures (AP# CIBR-IACUC-037).

### Surgical procedures
Mice were anesthetized with isofluorane (5% for induction and 1–1.5% for maintenance) and applied with an eye ointment to prevent dryness of the eyes. After disinfection with 75% v/v alcohol, the skull was exposed and leveled by adjusting the nose bar and two ear bars. For the CA1 imaging, a craniotomy was performed above the right dorsal CA1, with a 1.0–1.2 mm diameter and centered at AP −2.3 mm, ML 2.0 mm. Overlying cortical tissue was aspirated using a blunted 27 g and 28 g needle connected to a vacuum while under irrigation with sterile and cold saline until the corpus callosum became visible. Then, a blunted 30 g needle was used to remove the large superficial fibers, leaving a thin fiber layer intact. Sterile gelfoam was applied to stop bleeding. The 350 nL of GCaMP6m viral vectors (AAV2/9-hSyn-GCaMP6m-WPRE-pA, titer: -2.2 × 10^{12} gc/ml, Taitool Bioscience, Shanghai) was injected into the dorsal CA1 of C57BL/6 J mice at the coordinates of AP −2.3 mm, ML 2.0 mm, DV −1.3 mm (from bregma) at 0.8 nL/sec. After the virus injection, the infusion needle was slowly removed. A GRIN lens (GRINtech, 850 µm in diameter and 6.7 mm in length or GoFoton, 1 mm in diameter and 4 mm in length) was lowered to the tissue surface and then 50 µm deeper to reposition the swelling brain tissue. The cranial window was 1.2 mm in diameter for OFC imaging, centered at AP 2.5 mm and ML 1.5 mm. The overlying brain tissue was aspirated to a depth of -1.0 mm from the brain surface. After no visible bleeding, a GRIN lens was slowly implanted to a depth of 2.5 mm from bregma. Dental cement (C&B Metabond) was applied to fix the lens to the skull and attach a titanium headplate. Three weeks after virus injection and lens implantation, animals were head-restrained and examined for fluorescence signal under a miniscope (UCLA Miniscope V4) attached with a baseplate. The field of view and the distance between the objective and the lens were adjusted until the blood vessels and putative cell bodies were visible in the middle of the Miniscope adjustable focus plane range. Then, the baseplate was cemented to the animal's skull, and the miniscope was replaced with a 3D-printed cover to protect the lens. At least three days after baseplate attachment, the animals were water-restricted and handled daily before behavioral procedures.

### Histology
At the end of the behavioral and imaging experiment, mice were euthanized and then perfused with saline, followed by 4% paraformaldehyde (PFA) in phosphate-buffered saline (PBS, pH -7.4). The brain tissue was then carefully removed, post-fixated in 4% PFA for at least 6 h at room temperature, and dehydrated in 30% sucrose solution until the tissue sank (normally after 12 h at room temperature). After embedding and freezing, the brain tissue was then sectioned (50 µm) coronally using a cryostat microtome (CM3050 S, Leica) and mounted on slides with DAPI. The fluorescence signals (GCaMP6 and DAPI) were examined with a slide scanner (VS120, Olympus).

### Behavioral task
Mice were trained to perform two odor-reward association problems (Problem 1 and Problem 2). After acclimating to the head fixation and a treadmill, head-restrained mice were presented with a vacuum-controlled air-flow tube and a stainless-steel waterspout to deliver odors and sucrose solutions, respectively. The waterspout was connected to a tongue lick detector. Odor and reward deliveries were controlled by an Arduino board (Mega2560), and the task and behavioral signals were collected with a USB data acquisition card (6501, National Instruments, Texas).

In the training of Problem 1 (Day 1–12), mice learned four odor-reward sequences organized as below:

$$1a+: \text{Odor 1} \rightarrow \text{Odor 3} \rightarrow (+)$$

$$1a-: \text{Odor 2} \rightarrow \text{Odor 3} \rightarrow (-)$$

$$1b+: \text{Odor 4} \rightarrow \text{Delay} \rightarrow (+)$$

$$1b-: \text{Odor 5} \rightarrow \text{Delay} \rightarrow (-)$$

The + and − symbols indicate outcomes with and without sucrose delivery (5% w/v; ~5 μL). The four trial types were presented pseudo-randomly and independently for each day and mouse. The delivery of each odor lasted for 1 s. In the sequences with two odors (i.e., 1a+ and 1a−), the delay between the offset of the first odor and the onset of the second odor was 3 s, and the delay between the offset of the second odor and the outcome was 2 s; in the sequences with single odors (1b+ and 1b−), the delay between the offset of the odor and the outcome was 6 s. Trials of each trial type appeared 40 times per session and were separated by an inter-trial interval (ITI), sampled from a uniform distribution of 15–25 s.

In the training of Problem 2 (Day 13–17), procedures were identical to those in Problem 1, but with a different set of odors, i.e., Odors 6–10 replaced Odors 1–5:

$$2a+: Odor\ 6 \rightarrow Odor\ 8 \rightarrow (+)$$

$$2a-: Odor\ 7 \rightarrow Odor\ 8 \rightarrow (-)$$

$$2b+: Odor\ 9 \rightarrow Delay \rightarrow (+)$$

$$2b-: Odor\ 10 \rightarrow Delay \rightarrow (-)$$

In the final training of Problem 1 and Problem 2 (Day 18–29), eight sequences (i.e., 1a−, 1b+, 1b−, 2a+, 2a−, 2b+, and 2b−) from the previous two phases were combined and presented pseudo-randomly.

### Calcium imaging

To collect calcium imaging data, we mounted the head-restrained mice with a miniscope (UCLA Miniscope V4; resolution: 600 pixels × 600 pixels; sampling rate: 20 Hz; www.miniscope.org) after their habituation to the experimental environment for each session. Imaging parameters, including optical focus, LED intensity, and gain, were adjusted for each mouse to acquire optimal images. However, within each mouse, the parameters rarely varied across days since the baseplate fixed the field of view of each mouse. Videos were downsampled to 300 pixels × 300 pixels and 10 Hz and corrected for motion using the ImageJ (NIH, Bethesda). After data preprocessing, putative cells and their fluorescent traces were extracted using a constrained non-negative matrix factorization for the microendoscopic data algorithm (CNMF-E)[62]. Each cell was manually assessed based on calcium dynamics and spatial configuration, and cells with long decay time or silent activities (i.e., less than five calcium events on a given session) were excluded. The fluorescence traces of all remaining cells were deconvoluted to estimate spike rates using the OASIS algorithm embedded in the CNMF-E[62]. The raw fluorescence traces were z-scored to the average of the whole session for analysis using fluorescence signals. For longitudinal analyses, we applied the CellReg toolbox[63] to track the same group of cells across days, using spatial footprints of putative neurons extracted by the CNMF-E. Only neurons existing in all aligned sessions and with a cell score > 0.7 were considered successfully registered and used for further analyses.

### Behavioral analysis

To quantify the learned associative strength between odors and outcomes during learning, we calculated a behavioral performance index using the differential lick rates (LRs) at the time of 0–7 s (i.e., before outcome) between the rewarded (+) and non-rewarded (−) sequences in each sequence pair (i.e., 1a+ vs. 1a−, 1b+ vs. 1b−, 2a+ vs.

2a−, 2b+ vs. 2b−):

$$Performance\ index\ (PI) = \frac{LR(+) - LR(-)}{LR(+) + LR(-)} \quad (1)$$

The PI > 0 indicates a preference for rewarded sequences, hence more learning of cue-outcome associations.

### Splitting neuron identification and splitting strength quantification

In our non-spatial associative learning problems, a sequence-splitting neuron is a cell that exhibited different responses to the overlapping epochs that shared the same odor, delay, or reward in different sequences. To avoid the influence of fluorescence decay from previous calcium events, we used estimated spike rates by non-negative deconvolution for analyses, including calculating selectivity to paired sequences and identifying splitting neurons. Specifically, we binned estimated spikes every 1 s to calculate the sequence-selective cell fraction; then, we performed the Wilcoxon rank sum test to determine the statistical significance of each cell's differential firing rates to paired sequences (*$p < 0.01$). To identify splitting neurons of an overlapping epoch (odors in 1a+ vs. 1a− or 2a+ vs. 2a− at 4–7 s, delays in 1b+ vs. 1b− or 2b+ vs. 2b− at 4–7 s, and rewards in 1a+ vs. 1b+ or 2a+ vs. 2b+ at 7–10 s), we averaged the 3-second activities for each neuron, and a splitting neuron was identified when its differential responses in the two sequences passed the statistic test of Wilcoxon rank sum test (*$p < 0.01$). We used Spearman's rank correlation to test the correlations between the training days and fractions of sequence-splitting neurons.

To quantify the splitting strength of each neuron, we compared the distributions of its neural responses across trials to paired sequences by computing the area under the ROC (receiver operating characteristics) curve (AUC). The splitting strength of individual neurons was defined as the discrimination index (DI):

$$DI = |AUC - 0.5| \times 2 \quad (2)$$

Pairwise comparisons between the DI cumulative distributions of any two training days were conducted using the Kolmogorov-Smirnov test (*$p < 0.01$).

### Decoding analysis

We trained support vector machine (MATLAB toolbox LIBSVM-3.22 and ndt 1.0.4)[64] for binary decoding of paired sequences (1a+ vs. 1a−, 1b+ vs. 1b−, 2a+ vs. 2a−, 2b+ vs. 2b−) with the leave-one-out cross-validation approach to assess the accuracy. For the pseudo-ensemble decoding, neurons were randomly resampled 500 times from a pool of neurons imaged from all mice on each day. The real-ensemble decoding was performed on all neurons imaged from each session. The within-trial-type shuffling was performed for each neuron at each time point. After each shuffling, one trial from each trial type was selected as test data and the rest as training data. The classification accuracy obtained from 50 times cross-validation was averaged as the overall performance of this subset of neurons. The statistical significance of decoding accuracy was tested using a 95% CI determined by the same decoding procedure but using label-shuffled data.

### Coding similarity

We measured the similarity of two splitting signals based on their angle in the neural activity space[65]. For binary decoding of each sequence pair, e.g., 1a+ vs. 1a−, an SVM linear classifier was trained to define a hyperplane with the optimal separation, and the weight vector orthogonal to the hyperplane was the coding direction of the given sequence pair, also referred to as the splitting vector. Intuitively, the classification accuracy in generalizing to new splitting problems would

likely increase when their splitting vectors tended to be parallel. Therefore, we computed the cosine of the angle ($\theta$) between two splitting vectors ($\mathbf{s}_1$ and $\mathbf{s}_2$) to quantify their coding similarity (i.e., cosine similarity):

$$\cos\theta = \frac{\mathbf{s}_1 \cdot \mathbf{s}_2}{||\mathbf{s}_1|| \cdot ||\mathbf{s}_2||} \tag{3}$$

For each repeat, trials of each sequence pair were randomly divided into two halves, giving eight splitting vectors in total. The angle between two vectors from the same sequence pair was used to compute the coding similarity within the same splitting signals, with the cosine value closer to 1 suggesting less noise from trial-to-trial variability provided that the decoding performance was robust. The angle between two splitting vectors from different sequence pairs reflected their similarity, and a higher cosine value indicated more similar coding directions for the two splitting signals. Trials were re-divided 100 times. The statistical significance was tested using a 95% CI determined by the same analyses using label-shuffled data.

### Multidimensional scaling
We performed the MDS (*cmdscale*, MATLAB function) to visualize the geometry of neural representations of splitting signals. Specifically, all neurons recorded from a specific region, CA1 or OFC, in all mice were combined to create a pseudo-ensemble. Then we computed the Euclidean distance between two of the trial types during overlapping epochs (4–7 s) using trial-averaged data, used the distance matrix to perform MDS and visualized the results on the first two dimensions.

### Computational modeling
To quantify how the generalization of splitting signals was affected by other task variables, including the past and current cues ($\mathbf{a}$ and $\mathbf{b}$), sequence structure ($\mathbf{t}$), and expected outcome ($\mathbf{o}$), we built a linear model to integrate and estimate their contributions to the coding similarity between splitting vectors ($\mathbf{s}$) in each brain region, formalized as:

$$\mathbf{s}_i = \alpha \cdot \mathbf{a}_i + \beta \cdot \mathbf{b}_i + \gamma \cdot \mathbf{t}_i + \delta \cdot \mathbf{o}_i \tag{4}$$

$$\text{Coding similarity } (i,j) = \frac{\mathbf{s}_i \cdot \mathbf{s}_j}{||\mathbf{s}_i|| \cdot ||\mathbf{s}_j||} \tag{5}$$

where $i$ and $j$ indicate the $i^{th}$ and $j^{th}$ sequence pair, 1: 1a+ vs. 1a–; 2: 2a+ vs. 2a–; 3: 1b+ vs. b–; 4: 2b+ vs. 2b–. $\mathbf{a}_i$, $\mathbf{b}_i$, $\mathbf{t}_i$, $\mathbf{o}_i$ were zero-mean orthonormal (mutually orthogonal and with an L2 norm of 1) vectors representing the coding directions of the past cue, current cue or delay, sequence structure, and expected outcome of the $i^{th}$ sequence pair in the neural activity space. Specifically, $\mathbf{a}_i$, $\mathbf{b}_i$, $\mathbf{t}_i$, $\mathbf{o}_i$ were vectors with 100 random numbers drawn from normal distributions and then normalized by their L2 norms. $\mathbf{a}_1$, $\mathbf{a}_2$, $\mathbf{a}_3$, $\mathbf{a}_4$, $\mathbf{b}_1$, and $\mathbf{b}_2$ were uncorrelated ($r = 0$), corresponding to past ($\mathbf{a}_1$, $\mathbf{a}_2$, $\mathbf{a}_3$, $\mathbf{a}_4$) and current ($\mathbf{b}_1$, $\mathbf{b}_2$) odor cues. $\mathbf{b}_3$ and $\mathbf{b}_4$ were vectors of zeros indicating pure delay time in their corresponding paired sequences. $\mathbf{t}_1$ and $\mathbf{t}_2$ were highly correlated ($r = 1$); $\mathbf{t}_3$ and $\mathbf{t}_4$ were also highly correlated ($r = 1$) because of shared sequence structures. The vectors $\mathbf{o}_1$, $\mathbf{o}_2$, $\mathbf{o}_3$, $\mathbf{o}_4$ were highly correlated ($r = 1$) because of the same outcome predictions in all four paired sequences. Each splitting vector, $\mathbf{s}_i$ ($i = 1, 2, 3, 4$), was a weighted linear summation of four basis vectors ($\mathbf{a}_i$, $\mathbf{b}_i$, $\mathbf{t}_i$, $\mathbf{o}_i$), with $\alpha$, $\beta$, $\gamma$, $\delta$ their corresponding weights ranging from 0 to 1. Because the measurement of coding similarity only depended on the direction of splitting vectors but not their magnitude, we normalized $\mathbf{s}_i$ to unit length by dividing the weight vector ($\alpha$, $\beta$, $\gamma$, $\delta$) by its norm. Finally, the coding similarity between the $i^{th}$ and $j^{th}$ splitting vectors was computed as the cosine of the angle between them.

### Model simulation
To visualize the influence of each variable, we simulated four patterns of coding similarity by assigning a high value (0.98) to one parameter and a low value (0.11) to the other three in each condition. For example, to estimate the influence of past cue on the splitting signals, we set the parameters as $\alpha = 0.98$, $\beta = 0.11$, $\gamma = 0.11$, $\delta = 0.11$.

### Model fitting
We fit the model with $\mathbf{a}$, $\mathbf{b}$, $\mathbf{t}$, $\mathbf{o}$ vectors generated as above together with real data of coding similarity in CA1 and OFC, resulting in the optimal weight vectors ($\alpha$, $\beta$, $\gamma$, $\delta$) that estimate the influence of each variable in splitting signal generalization. Specifically, we conducted function optimization (*fminsearchbnd*, MathWorks File Exchange) with the simplex search method of Lagarias. We used the mean squared error (MSE), as the cost function, with the maximum number of function evaluations allowed being 20,000. The weight vector from each search was normalized by its L2 norm. Before MSE calculation, the predicted coding similarity was multiplied by a noise term, computed as the average of the within-splitting signal coding similarity of four sequence pairs, considering trial-to-trial variability in real data. For evaluating each set of parameters, the predicted coding similarity was the average of 5000 coding similarity values, each with differently generated $\mathbf{a}$, $\mathbf{b}$, $\mathbf{t}$, $\mathbf{o}$ vectors. Model fitting in each brain region was performed 100 times, resulting in 100 sets of recovered parameters. To quantify the difference in the impact of four task variables in CA1 and OFC, we used two-way ANOVA to examine each parameter, with two factors being the 'brain region' and 'training day'.

### Reporting summary
Further information on research design is available in the Nature Portfolio Reporting Summary linked to this article.

## Data availability
The dataset used in this study has been deposited to a public repository at https://doi.org/10.17605/OSF.IO/YSNQT. Source data are provided with this paper.

## Code availability
The MATLAB scripts used in this study has been deposited to a public repository at https://doi.org/10.17605/OSF.IO/YSNQT.

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

## Acknowledgements

This work was supported by the STI 2030-Major Projects (2022ZD0207500 to J.Z.), Beijing Nova Program (Z211100002121029 to J.Z.), and Chinese Institute for Brain Research (CIBR), Beijing. The authors thank the services provided by the Laboratory Animal Resource Center, Optical Imaging Facility, and Instrumental Core at CIBR; Zhongyang Qi (NIBS) for technical assistance with calcium imaging; Ni Ji (CIBR) and Tatsuo Okubo (CIBR) for discussions on data analyses and computational modeling; Yingjun Tang (CIBR) for the help with figure illustrations; Geoffrey Schoenbaum (NIH) for helpful comments on an earlier version of the manuscript.

## Author contributions

H.L. and J.Z. designed the experiments. H.L. and J.Z. built the behavioral and imaging setups. H.L. collected and analyzed the data, with advice and assistance from J.Z. H.L. and J.Z. wrote the manuscript. J.Z. supervised the project.

## Competing interests

The authors declare no competing interests.
