## [Peer Review File · Nature Communications]

Hippocampal and orbitofrontal neurons contribute to complementary aspects of associative structureReviewer #1 (Remarks to the Author):

In this study, Lin & Zhou performed calcium imaging in the hippocampal CA1 region and the prefrontal OFC using GRIN lenses for several days during an olfactory guided discrimination task. The behavioral task was quite complex and characterized by two distinct odor-related reward associations. Parts of the behavioral task design were characterized by different sets of odors but similar task structure. Some of the odor sequences were paired with a reward outcome whereas others not. The question addressed in the study is whether hippocampal and prefrontal cells encode associative information differentially. The main finding is that the OFC cells shared between sequence pairs with the same task structure and thus encode more schema representations whereas the hippocampus encoded more proximal olfactory cues, thus the temporal context. Although there are some interesting findings particularly based on the support vector machine (SVM) decoder that indicated a stronger generalization between common task structure in the OFC compared to the hippocampus, the experimental findings are not that novel.

Major concerns:

1. In general: the wordings: experimental problem 1 and 2, odor sequences a and b, odor Cue A and B and odor 1-8, combined with the hardly to differentiate color code 'blue and green' makes reading and following the experimental design a quite challenging endeavor.

2. The prefrontal cortex and hippocampus receive direct olfactory inputs. Based on the recordings shown in Figure 2, and based on previous work, it appears that cells are tuned to olfactory stimuli, either to the first (delay 0 sec), the second odor (delay 4 sec), to both or to the outcome namely the reward (delay 7 sec). Previous work showed that olfactory tuning can improve across learning on subsequent days as well as the differentiation between olfactory cues (Wood et al., Neuron 2020). Tuning and olfactory representation in the hippocampus encodes for future behavior (reward obtainment and anticipatory licking or not; Woods et al., 2020). Similarly, olfactory tuning can be observed in this study (Figure 2) with the expected bias towards outcome encoding in the OFC (reward related activity). Thus, improved representation across learning and the behavioral outcome is expected based on previous literature as well as the differentiation of odor-related rewards.

Since only one of the two related combinations S1a+ vs S1a- and S1b+ vs S1b- are rewarded, it is expected that the motivation to discriminate is high. In contrast, the discrimination between S1a+ and S1b+ is not needed for the animal because it learned that both are rewarded. In summary, the olfactory tuning is an important encoding of information for reward-related behavior, and that is shown with the original data in Figure 2 and 3. Therefore, the wording 'splitting cells' is misleading.

The ability of the OFC to generalize between common task structure is interesting.

In summary, the data should be published but the experimental design, its description in the results section and the color code should be improved readability.

Minor:

1. The conclusions why task sequence pair S2b+ vs. S2b- was learned more rapidly than they learned S1b+ vs. S1b-, but that task sequence S2a+ vs. S2a- was not learned faster or slower when compared to S1a+ vs. S1a- is very speculative at this part of the results section. There are no evidence for this speculation given.

2. Figure 1b: color coded licks should be combined with the odor definition (1-8). The color blue and green can be only hardly be discriminated.

3. It remains unclear why the authors injected AAVs encoding GCAMP6m and combined it with imaging Thy1-mice expressing GCAMP6f; why not sticking to the same GCAMP with similar binding and unbinding kinetics for calcium?

4. time axis in Figure 2 and 3, in particular the meaning of the vertical dashed lines need further explanation. Its relationship to odor A and B and reward onset should be mentioned.

Reviewer #2 (Remarks to the Author):

The manuscript by Lin and Zhou combines in vivo calcium imaging, behavior, and computational modeling to attempt to identify unique contributions of hippocampal CA1 and orbitofrontal cortex (OFC) to the encoding of odor-outcome sequence pairs. They varied temporal information, task structure, and predicted outcome (i.e., features) using the different odor-outcome sequence pairs. They propose that while both CA1 and OFC show neural ensembles encoding the different features, CA1 plays a greater role in the temporal information context, whereas OFC plays a greater role in representing task structure and predicting outcomes. I thought that the study was well-designed, appropriate control epochs were used, and the findings make important observations for the field. I have some suggestions to further strengthen the study (in the order that they appear):

1. Why were different GCaMP6 variants used for CA1 vs. OFC? Please explain and clearly alert the reader (~line 11) of this difference in their preparations.
2. Related to Figs 2-3: Within the 3-s time windows used for in-depth analysis, what fraction of neurons showed activation in different temporal phases (e.g., start vs. end of the time window)? This can be observed in the representative plots in Fig. 2b, left column (odor-splitting cells). Please plot this for CA1 and OFC, for the different splitting neurons.
3. Related to Figs 2-3: Stability of the splitting cell populations should be examined more thoroughly (in addition to some data shown in Fig. 6a, d). Specifically, were the cells that were splitting for the different features early on in their task, stable in later stages? For instance, in Fig. 2b, left column, day 1 cell 46 already displays splitting; using longitudinal registration, how stable are these splitting cells? Do this for both cells arising early on in training as well as others that show up later in the task. Depending on when splitting was detected for a cell, early vs. later, are there any stability differences? Further, for CA1 vs. OFC, for a given splitting population, was there a difference in how many cells were added to this population across learning (i.e., cells dropping into a given splitting function)? These analyses may reveal further differences between these two important brain regions.
4. Related to Figs 2-3: Although the presence of learning-related splitting cell populations in both CA1 and OFC supports their role in this behavior, performing loss of function manipulations (e.g., optogenetic inhibition experiments) to directly demonstrate that individually CA1 and OFC are required for the sequence discrimination behavioral paradigm used in this study will increase the significance of these neural findings.
5. Related to Fig. 3c, d: Do odor-splitting and delay-splitting neuronal populations increase in size across learning to a greater extent in OFC as compared to CA1? Quantifying and comparing this aspect between CA1 and OFC would be informative.
6. Line 142: For CA1, reward-splitting cells while not significant, still shows a strong trend. For this reason, update line 142 to clearly highlight this point and talk about the potential relevance of this being different between CA1 and OFC.
7. Related to Fig 3: Can the authors examine whether odor splitting cells ever become delay or reward splitting (potentially across days) and vice versa? This may reveal plasticity in these splitting populations and is something that would be interesting to compare between CA1 and OFC.
8. Line 147: Any significance of odor splitting cells showing significant DI over time in both brain regions? Please alert readers that delay splitting cells showed a similar trend but did not reach significance.
9. Line 162: How did they select 80 neurons? Please compare these results to different ranges of neuronal numbers to determine the robustness of their observations. This may also help identify a potential minimum neuronal number needed for comparable results, which would be very informative.
10. Line 238: What about reward-splitting cells between problems? This is relevant given the data in Fig. 3c (right graph) for CA1.

Reviewer #3 (Remarks to the Author):

The goal of this study is to identify the distinct contributions of hippocampus (HC) and orbitofrontal

cortex (OF) to associative memory. Although these two brain structures have been extensively studied (particularly the HC), it can be difficult to directly compare their coding properties because behavioral paradigms have varied between studies. In this study, the authors recorded neural activity in HC or OF in separate groups of animals performing the same odor tasks. Overall, this is an important and timely problem to address (but see below).

The main strengths of the paper are on the technical side. Recording neural activity in such complex behavioral paradigms is a significant technical challenge and the authors should be commended for it. The data analysis is also extensive and (mostly) well presented in figures that are visually compelling. Clearly, a significant amount of work and technical skill went into this manuscript.

However, the manuscript lacks sophistication at the conceptual level but also in terms of overall clarity. This is not simply a need for polishing the language a bit more – there is some major rethinking and rewriting to do. My main concerns are outlined below.

MAIN CONCERNS

1. There is a major disconnect between the high-level description of the results (in abstract, introduction, and discussion) and the results presented (in results section and figures). The high-level description presents complementary roles for HC and OF, with HC showing “modulation based on temporal context” and OF representing “common task structure and outcome prediction”. In contrast, the results and figures rely heavily on the “sequence-splitting” terminology (odor-, delay-, and reward-splitting neurons) and the connection between the two is hard to follow. I’m still not sure how their HC results have anything to do with “temporal context”. Even “task structure” seems to have different meanings in different parts of the manuscript. While I appreciate the effort to make the abstract and introduction more appealing to a broader audience, there needs to be a better connection with the results that are actually presented. For example, statements to the effect that HC and OF “complementarily organize associative memory” (abstract) or are responsible for “distinct components of associative memory” (discussion) and not well supported by the presented data. There are many more such examples.

2. The novelty of the findings is debatable.

The rationale for the experiment can be paraphrased as “we already know a lot about the coding of HC and OF neurons, but studies comparing the two in the same behavioral task are rare”. That is technically true, but it does not make this an important study. Implied here is that this study would reveal differences between HC and OF that were not already known, and it does not. Also implied is that this would be a direct comparison between HC and OF (i.e., activity recorded in both structures in the same animals), and this is not the case. To be fair, the authors do mention that the data from each region come from separate groups of animals (on line 93 and in the Supplementary Table), but such key information should come earlier and be clearer.

3. Clarity is a major problem with this manuscript.

I am an expert in this area, and I had to read the manuscript several times to understand it. Captions that are more narrative in nature (explaining, not just listing, what is shown) would help tremendously. A consistent and simpler terminology for the different trial types would also be important. The current terminology (S1a+, S1a-, S1b+, S1b-, S2a+, ...) is OK when there aren't too many comparisons to track (up to Fig 3) but it becomes overwhelming after that. This applies to the main text as well, since those trial types are used as examples of the contrasts of interest. I can't imagine a non-expert would be able to follow the manuscript in its current form.

4. The use of pseudo-ensembles (e.g., Fig 4) is questionable.

The authors recorded enough cells per animal to decode ensembles properties in each animal individually and report central tendencies at the group level. I understand that some animals had fewer cells and that some “lost” their cells at some point during testing, but there are a number of ways to aggregate data in such cases (even with basic GLMs). Pseudo-ensembles should be viewed as a last resort, as for a preparation yielding only a few cells per animals.

5. The results presented in Figures 5, 6 and 7 are very complex but seem to add little to the story.

If I understand correctly, Figs 2 and 3 show similar coding properties between HC and OF. Differences between HC and OF appear in Fig 4 (panels e vs f) which shows that the black curve is shifted up for OF, reflecting more similarity between odor-splitting and delay-splitting vectors (the implication is that the coding in HC is more conjunctive). As far as I know, that's the main finding. Figures 5-7 seem to be different ways to show the same thing with more complex models -- their added value is unclear.

MINOR COMMENTS

1. To me, the discussion section read as overly speculative and broad given the findings. I also felt it should include a discussion of the limitations of the study.

2. In Fig 1, performance of the HC and OF groups should be shown separately.

3. I understand that the cell counts per animal are shown in Supplementary Table 1, but it is customary to include a basic description of the counts (total and mean per animal or region) in the main text.

4. Why show variability as standard deviation for 500 runs (e.g., Fig 4)? Why not confidence intervals for the curve themselves?

5. On that note, showing the confidence intervals for shuffled data as a line (instead of an interval) is a bit counterintuitive. Does it reflect the upper bound of the estimate (mean + CI95%)? This should be clarified.

6. It's unclear what the blue and green traces are in Fig 4e,f.

7. This part of OF appears very anterior. Is there a reason this specific segment was targeted? Does it match that of previous studies?

8. The imaging procedure resulted in considerable brain damage above the region of interest (particularly in OF; Suppl Fig 2), which I would recommend disclosing. Perhaps it would help reassure the reader to clarify that similar coding properties have been reported with electrophysiology (so the brain damage isn't a major concern)?

9. A couple of papers by Ramus are particularly relevant to this paper:

Ramus & Eichenbaum (2000) Neural correlates of olfactory recognition memory in the rat orbitofrontal cortex. *J Neurosci*: 20, 8199-208. <https://doi.org/10.1523/jneurosci.20-21-08199.2000>

Ginther MR, Walsh DF, Ramus SJ (2011) Hippocampal neurons encode different episodes in an overlapping sequence of odors task. *J Neurosci* : 31(7), 2706-11. doi: 10.1523/JNEUROSCI.3413-10.2011.

REVIEWER COMMENTS

Reviewer #1 (Remarks to the Author):

In this study, Lin & Zhou performed calcium imaging in the hippocampal CA1 region and the prefrontal OFC using GRIN lenses for several days during an olfactory guided discrimination task. The behavioral task was quite complex and characterized by two distinct odor-related reward associations. Parts of the behavioral task design were characterized by different sets of odors but similar task structure. Some of the odor sequences were paired with a reward outcome whereas others not. The question addressed in the study is whether hippocampal and prefrontal cells encode associative information differentially. The main finding is that the OFC cells shared between sequence pairs with the same task structure and thus encode more schema representations whereas the hippocampus encoded more proximal olfactory cues, thus the temporal context. Although there are some interesting findings particularly based on the support vector machine (SVM) decoder that indicated a stronger generalization between common task structure in the OFC compared to the hippocampus, the experimental findings are not that novel.

Thank you for your time reviewing our paper, recognizing our main findings as interesting, and providing detailed suggestions. After performing more experiments and rewriting, we believe the current revised manuscript has been substantially improved and, hopefully, with its novelty, better demonstrated. The novelty of this study lies not only in the methodology but also in some of the unexpected discoveries. For instance, as mentioned in the reviewer's above evaluation, it is quite a novel finding that there was a much stronger generalization between learned associations sharing a common task structure (i.e., task schema) in the OFC compared to the hippocampus. Overall, we believe these discoveries, both expected and unexpected, resulting from direct comparisons between the hippocampus and OFC in a non-spatial task, contribute to our understanding of the intricate processes underlying associative structure organization in the brain.

Major concerns:

1. In general: the wordings: experimental problem 1 and 2, odor sequences a and b, odor Cue A and B and odor 1-8, combined with the hardly to differentiate color code 'blue and green' makes reading and following the experimental design a quite challenging endeavor.

Thank you for raising this clarity issue. We have now thoroughly updated the figures and text to describe the task design and differentiate task conditions more clearly.

2. The prefrontal cortex and hippocampus receive direct olfactory inputs. Based on the recordings shown in Figure 2, and based on previous work, it appears that cells are tuned to olfactory stimuli, either to the first (delay 0 sec), the second odor (delay 4 sec), to both or to the outcome namely the reward (delay 7 sec). Previous work showed that olfactory tuning can improve across learning on subsequent days as well as the differentiation between olfactory cues (Wood et al., Neuron 2020). Tuning and olfactory representation in the hippocampus encodes for future behavior (reward obtainment and anticipatory licking or not; Woods et al., 2020). Similarly, olfactory tuning can be observed in this study (Figure 2) with the expected bias towards outcome encoding in the OFC (reward related activity). Thus, improved representation across learning and the behavioral outcome is expected based on previous literature as well as the differentiation of odor-related rewards.

The splitting signals cannot be explained by differential responses to the second odor stimuli because our analyses focused on the time when there was a common odor (e.g., 1a+ vs. 1a-) or a delay (without a common odor, e.g., 1b+ vs. 1b-) between paired sequences. However, the differential responses to the same odor or delay might be traced back to the first odor stimuli.

Thus, if we understood the comment correctly, the reviewer proposes that increased splitting signals might be due to improved olfactory tuning to the first odor cues—this is, however, not contradictory to our interpretation of the data because our exact goal was to explain the meaning of such splitting signals, specifically, whether these splitting signals reflected influence from the past sensory cues, expected outcome, or task schema shared between cue-outcome associations. There was evidence from the literature that these task variables modulate neural activities in both brain regions; however, the extent to which and the differences between the two brain regions are affected by these modulations were unclear, especially under the same experimental conditions. To investigate, we tested how these task variables differently affected splitting signal generalization and thus could help identify particular content in learned associations. We found that the splitting signals in the hippocampus were prominently influenced by a combination of past and current sensory cues; in contrast, those in the OFC were predominantly influenced by the expected outcome and task schema shared between similar cue-outcome associations.

Since only one of the two related combinations S1a+ vs S1a- and S1b+ vs S1b- are rewarded, it is expected that the motivation to discriminate is high. In contrast, the discrimination between S1a+ and S1b+ is not needed for the animal because it learned that both are rewarded. In summary, the olfactory tuning is an important encoding of information for reward-related behavior, and that is shown with the original data in Figure 2 and 3. Therefore, the wording 'splitting cells' is misleading. The ability of the OFC to generalize between common task structure is interesting.

As in our response to the above comment, we used the word 'splitting cells' to define neurons showing differential activities to paired sequences at the time of a common odor or delay. Such a signal would often be explained by outcome expectancy, but this was to be tested since it might encode past sensory cues, and maybe other task variables as well. In this study, our exact goal was to determine the meaning of such a 'splitting signal', which includes the past and current sensory cues, expected outcome, and task schema shared between cue-outcome associations. With the current task design, we were able to identify these components from these splitting signals, and the results showed that the two brain regions contributed differently to the splitting signals.

In summary, the data should be published but the experimental design, its description in the results section and the color code should be improved readability.

Thank you, and we really appreciate your support in publishing our results. We have thoroughly revised the manuscript based on your and other reviewers' invaluable comments.

Minor:

1. The conclusions why task sequence pair S2b+ vs. S2b- was learned more rapidly than they learned S1b+ vs. S1b-, but that task sequence S2a+ vs. S2a- was not learned faster or slower when compared

to S1a+ vs. S1a- is very speculative at this part of the results section. There are no evidence for this speculation given.

Yes, the reason was speculative since we do not have direct evidence. We have rephrased the description of the behavioral results (**lines 81-88**).

2. Figure 1b: color coded licks should be combined with the odor definition (1-8). The color blue and green can be only hardly be discriminated.

We have changed these colors to make different trial types more discriminable in the plots.

3. It remains unclear why the authors injected AAVs encoding GCaMP6m and combined it with imaging Thy1-mice expressing GCaMP6f; why not sticking to the same GCaMP with similar binding and unbinding kinetics for calcium?

We began the study by using AAV-GCaMP6m and later also included the GCaMP6f transgenic mice. Indeed, the two calcium indicators have different unbinding kinetics for calcium. However, the ascending phases of both signals, on which the deconvoluted or inferred spikes have depended (Friedrich et al., PLOS Comput Biol 2017; Kinsky et al., Nat Commun 2020), are almost identical (Chen et al., Nature, 2013), suggesting that the deconvoluted spikes from the calcium signals should mitigate or eliminate unbinding kinetic differences between calcium indicators—this was indeed the case based on our comparison of estimated spike rates between the two indicators (**Supplementary Fig. 18**). To further validate, we replicated our CA1 imaging experiments by collecting imaging data from four Thy1-GCaMP6f mice and compared the results with that using the GCaMP6m virus. Not surprisingly, the findings from the two cohorts of mice remained highly consistent regardless of specific GCaMP6 variants used. In the revised manuscript, we continued to present the data collected before and also placed the additional results in the **Supplementary Fig. 18**. Please note that prior to training in the current task, these four mice had been briefly trained with a simpler odor-reward association task in a piloting experiment, which might explain their more rapid learning and increases in splitting neurons during Problem 1. We also stopped imaging after they accomplished six days of training on Problems 1 and 2 (instead of 12 days in the initial experiment) since the data was already sufficient to meet our validation purpose. A brief description and explanation are added to the Results in the main text (**lines 89-93 and 254-259**). More details are in the caption of **Supplementary Fig. 18**.

Besides, in a separate project, using the same approach, we imaged CA1 neurons from a third cohort of Thy1-GCaMP6f mice ($n = 4$) performing the same task. These mice were naive on Day 1 of training but with unilateral DREADD inactivation of the OFC. The results were almost identical to the ones from the first cohort of mice that used AAV-GCaMP6m. Since this is still an ongoing project and we believe the above data has already been sufficient to demonstrate the robustness of the findings regardless of specific GCaMP6 variants being used, we did not include these newly collected data in the current manuscript, but they are available upon request.

4. time axis in Figure 2 and 3, in particular the meaning of the vertical dashed lines need further explanation. Its relationship to odor A and B and reward onset should be mentioned.

Done. Thank you!

Reviewer #2 (Remarks to the Author):

The manuscript by Lin and Zhou combines in vivo calcium imaging, behavior, and computational modeling to attempt to identify unique contributions of hippocampal CA1 and orbitofrontal cortex (OFC) to the encoding of odor-outcome sequence pairs. They varied temporal information, task structure, and predicted outcome (i.e., features) using the different odor-outcome sequence pairs. They propose that while both CA1 and OFC show neural ensembles encoding the different features, CA1 plays a greater role in the temporal information context, whereas OFC plays a greater role in representing task structure and predicting outcomes. I thought that the study was well-designed, appropriate control epochs were used, and the findings make important observations for the field. I have some suggestions to further strengthen the study (in the order that they appear):

1. Why were different GCaMP6 variants used for CA1 vs. OFC? Please explain and clearly alert the reader (~line 11) of this difference in their preparations.

As in the responses to Reviewer1, we began the study by using AAV-GCaMP6m and later also included the GCaMP6f transgenic mice. Indeed, the two calcium indicators have different unbinding kinetics for calcium. However, the ascending phases of both signals, on which the deconvoluted or inferred spikes have depended (Friedrich et al., PLOS Comput Biol 2017; Kinsky et al., Nat Commun 2020), are almost identical (Chen et al., Nature, 2013), suggesting that the deconvoluted spikes from the calcium signals should mitigate or eliminate unbinding kinetic differences between calcium indicators—this was indeed the case based on our comparison of estimated spike rates between the two indicators (**Supplementary Fig. 18**). To further validate, we replicated our CA1 imaging experiments by collecting imaging data from four Thy1-GCaMP6f mice and compared the results with that using the GCaMP6m virus. Not surprisingly, the findings from the two cohorts of mice remained highly consistent regardless of specific GCaMP6 variants used. In the revised manuscript, we continued to present the data collected before and also placed the additional results in the **Supplementary Fig. 18**. Please note that prior to training in the current task, these four mice had been trained briefly with a simpler odor-reward association task in a piloting experiment, which might explain their more rapid learning and increases in splitting neurons during Problem 1. We also stopped imaging after they accomplished six days of training on Problems 1 and 2 (instead of 12 days in the initial experiment) since the data was already sufficient to meet our validation purpose. A brief description and explanation are added to the Results in the main text (**lines 89-93 and 254-259**). More details are in the caption of **Supplementary Fig. 18**.

Besides, in a separate project, using the same approach, we imaged CA1 neurons from a third cohort of Thy1-GCaMP6f mice ($n = 4$) performing the same task. These mice were naive on Day 1 of training but with unilateral DREADD inactivation of the OFC. The results were almost identical to the ones from the first cohort of mice that used AAV-GCaMP6m. Since this is still an ongoing project and we believe the above data has already been sufficient to demonstrate the robustness of the findings regardless of specific GCaMP6 variants being used, we did not include these newly collected data in the current manuscript, but they are available upon request.

2. Related to Figs 2-3: Within the 3-s time windows used for in-depth analysis, what fraction of neurons showed activation in different temporal phases (e.g., start vs. end of the time window)? This can be observed in the representative plots in Fig. 2b, left column (odor-splitting cells). Please plot this for CA1 and OFC, for the different splitting neurons.

As requested, we plotted the fractions of neurons showing activation in different temporal phases for both CA1 and OFC (**Supplementary Fig. 3; lines 101-103**) with a 1-second temporal resolution, and did not observe significant differences in activation fractions at the start vs. end of the time window. Overall, there were increasingly more neurons recruited after learning, and there were generally more neurons activated in the OFC than in the CA1.

3. Related to Figs 2-3: Stability of the splitting cell populations should be examined more thoroughly (in addition to some data shown in Fig. 6a, d). Specifically, were the cells that were splitting for the different features early on in their task, stable in later stages? For instance, in Fig. 2b, left column, day 1 cell 46 already displays splitting; using longitudinal registration, how stable are these splitting cells? Do this for both cells arising early on in training as well as others that show up later in the task. Depending on when splitting was detected for a cell, early vs. later, are there any stability differences? Further, for CA1 vs. OFC, for a given splitting population, was there a difference in how many cells were added to this population across learning (i.e., cells dropping into a given splitting function)? These analyses may reveal further differences between these two important brain regions.

Thank you for the suggestion of doing more in-depth analyses on comparing the stability of splitting signals between the two brain regions. Through longitudinal registration, we tracked the activities of neurons throughout learning and indeed found that some splitting cells were dynamically changing across days. Specifically, some cells only showed sequence-splitting activities during late training ('adding'), and some cells exhibited splitting activities early during training but did not show such activities during late training ('dropping'). We quantified such dynamic changes for both brain regions through learning. Overall, we found that splitting cells in the CA1 changed more rapidly than those in the OFC, and in both regions, splitting cells became more stable during the late phase of training. We added these analyses to the **Supplementary Fig. 7-11** and the text to the Results (**lines 138-149**). However, we did not put too much emphasis on these results due to the considerations below.

Firstly, while we understand the significance of carrying out longitudinal analyses and highly appreciate that the reviewer has pointed out this important issue, the central question raised in the current study, however, can be well-addressed regardless of knowing the stability of splitting cells across days. We had a concern that putting too much emphasis on longitudinal analyses would distract readers' attention from these main findings. Secondly, there existed drawbacks due to the use of single-photon imaging and the fact that the focal plane was not intentionally maintained constant but slightly adjusted for the best imaging across days, although we are confident that the current longitudinal analyses were vigorous under our careful cell registration approach. Finally, in a separate project, we are specifically focusing on the longitudinal analyses by collecting imaging data from mice performing the same task but using two-photon imaging with improved quality of longitudinal tracking of the same neurons throughout training, which would enable much more detailed and comprehensive analyses and answer many questions beyond the scope of the current study.

We hope the reviewer will agree and find this response acceptable. Otherwise, we would be happy to adjust based on any further feedback from you.

4. Related to Figs 2-3: Although the presence of learning-related splitting cell populations in both CA1 and OFC supports their role in this behavior, performing loss of function manipulations (e.g., optogenetic inhibition experiments) to directly demonstrate that individually CA1 and OFC are required for the sequence discrimination behavioral paradigm used in this study will increase the significance of these neural findings.

We agree with the reviewer on the importance of investigating the behavioral significance of neural correlates. In this Pavlovian task, we used anticipatory licking as behavioral evidence showing mice have learned the association between the cue and outcome. Unlike operant behaviors, such anticipatory licking is not mandatory and thus only reflective of mice's appreciation of the associative structure available between the cue and outcome. In our opinion, anticipatory licking is only a rough behavioral measurement to show that mice have learned to predict the upcoming reward based on sensory cues but fall short of providing more specific information regarding higher-level cognitive processes like generalization—this is evident in our current dataset.

Firstly, during learning Problem 1, the mice's behavioral performance index based on anticipatory licking reached asymptotes on Day 5; however, their neural splitting signals continued to increase throughout 12 days of learning. Secondly, the activities of most neurons in the OFC and CA1 did not show a direct correlation with the licking rate or duration *per se*. Thirdly, strong splitting signals emerged even in mice that did not show much anticipatory licking to the expected reward outcome. Lastly and most importantly, the generalization between splitting signals, which is the focus of our current study, can only be read out through neural signals but not licking behaviors.

Although we agree with the reviewer on the significance of loss-of-function studies, due to these considerations, we decided not to do the perturbation in the two brain regions to just look at mice's licking behavioral changes in this study. Instead, we believe a more appropriate approach would be to directly examine neural generalization in one brain region while inactivating inputs from another brain region to study the interdependence of the two brain regions in the observed neural representations regarding formed associative structure. We have started to undertake such a project but realized the workload would greatly exceed the current study due to the requirement of many control groups.

In sum, meanwhile, future loss-of-function studies would unquestionably provide a more comprehensive and fuller understanding of the interaction between the two brain regions; we strongly believe that the current study has already delivered sufficient evidence, thus significant enough, to show brain-region differences in representing different aspects of associative structure. Again, we would be happy to hear any further comments from you on this issue and revise accordingly.

5. Related to Fig. 3c, d: Do odor-splitting and delay-splitting neuronal populations increase in size across learning to a greater extent in OFC as compared to CA1? Quantifying and comparing this aspect between CA1 and OFC would be informative.

Yes, indeed, the odor- and delay-splitting neuronal populations increased to a greater extent in the OFC than in the CA1. We have quantified their differences throughout learning and added the statistical

analysis to the Results (odor-splitting: group: $F(1, 108) = 30.34$, $***p < 0.0001$, day: $F(11, 108) = 6.82$, $***p < 0.0001$; delay-splitting: group: $F(1, 108) = 278.01$, $***p < 0.0001$, day: $F(11, 108) = 7.49$, $***p < 0.0001$, two-way ANOVA; **lines 115-122**).

6. Line 142: For CA1, reward-splitting cells while not significant, still shows a strong trend. For this reason, update line 142 to clearly highlight this point and talk about the potential relevance of this being different between CA1 and OFC.

Thank you for the suggestion. Such a trend ($p = 0.08$, Spearman's correlation) in the CA1 but not in the OFC ($p = 0.59$, Spearman's correlation) is suggestive of more generalization in the OFC due to the shared expected outcome, which is consistent with later main analyses that focused on odor- and delay-splitting signals. We now added the text to the Results (**lines 122-126**).

7. Related to Fig 3: Can the authors examine whether odor splitting cells ever become delay or reward splitting (potentially across days) and vice versa? This may reveal plasticity in these splitting populations and is something that would be interesting to compare between CA1 and OFC.

As in our response to an earlier question, we performed longitudinal tracking of the same neurons throughout mice learning Problem 1 and the two problems together and found splitting neurons were not static across days but dynamically changing. More specific to this particular question, in both brain regions, odor-, delay-, and reward-splitting cells were interchangeable across days. A detailed quantification has been presented in the **Supplementary Fig. 8 and 11 and lines 146-149**.

8. Line 147: Any significance of odor splitting cells showing significant DI over time in both brain regions? Please alert readers that delay splitting cells showed a similar trend but did not reach significance.

Thanks for pointing out these issues. There were indeed significant differences of neurons' DIs in the CA1 and OFC in terms of both odor- and delay- splitting signals. Apologies for not making it clear that the DI trends for the delay-splitting signals in both brain regions did not reach significance. We have now revised the text accordingly (**lines 131-132**).

9. Line 162: How did they select 80 neurons? Please compare these results to different ranges of neuronal numbers to determine the robustness of their observations. This may also help identify a potential minimum neuronal number needed for comparable results, which would be very informative.

We tried different neuronal numbers for decoding analyses and found 80 to be representative, such that the decoding accuracy did not reach an asymptote (100%). In accordance with the reviewer's suggestion and to better illustrate the decoding results for more transparent comparisons, we plotted the decoding results for three representative days (Day 1, 3, and 12) with different numbers of neurons randomly selected from 20 to all imaged neurons each day (**Fig. 4 and lines 155-166**).

Besides, beyond comparing how well neurons in the two brain regions could decode paired sequences, we wanted to make sure the decoding was optimal (i.e., close to 100%) with a sufficient number of neurons, which was a prerequisite for the following coding similarity analyses that used the decoding hyperplanes. The results showed that at least starting from Day 3, the decoding of any paired

sequences could reach almost optimal with only ~320 neurons in CA1 and ~160 neurons in OFC. Then, the following coding similarity analyses were appropriate since we used all imaged neurons.

10. Line 238: What about reward-splitting cells between problems? This is relevant given the data in Fig. 3c (right graph) for CA1.

Indeed, the results showing changes in fractions of reward-splitting neurons in **Fig. 3** were consistent with our main conclusion that splitting signals in the OFC were more generalized than those in the CA1. For instance, overall, the reward-splitting neurons appeared more in CA1 than OFC. However, because of the extremely low fractions of reward-splitting neurons (CA1: 4.97%, OFC: 2.21%, Problem 1 and 2 averaged; Problem1 and 2 overlapping fraction: CA1 3.60%, OFC 6.29%; $*p < 0.01$, rank sum test) compared to odor- and delay-splitting neurons in both brain regions, we decided not to focus on these neuronal populations. As introduced in the response to the last question, high decoding accuracy was required for the following coding similarity analysis. We were concerned that without sufficient fractions of reward-splitting neurons enabling high accuracy of decoding, the coding similarity analyses would be unreliable and might lead to misleading outcomes.

Reviewer #3 (Remarks to the Author):

The goal of this study is to identify the distinct contributions of hippocampus (HC) and orbitofrontal cortex (OF) to associative memory. Although these two brain structures have been extensively studied (particularly the HC), it can be difficult to directly compare their coding properties because behavioral paradigms have varied between studies. In this study, the authors recorded neural activity in HC or OF in separate groups of animals performing the same odor tasks. Overall, this is an important and timely problem to address (but see below).

The main strengths of the paper are on the technical side. Recording neural activity in such complex behavioral paradigms is a significant technical challenge and the authors should be commended for it. The data analysis is also extensive and (mostly) well presented in figures that are visually compelling. Clearly, a significant amount of work and technical skill went into this manuscript.

However, the manuscript lacks sophistication at the conceptual level but also in terms of overall clarity. This is not simply a need for polishing the language a bit more – there is some major rethinking and rewriting to do. My main concerns are outlined below.

Thank you for your time and appreciation of the importance of the problem we tried to address, the technical challenge, and our efforts put into this manuscript. We also would like to thank you for the precise criticism that encouraged us to rethink our results and revise the manuscript to make its significance better revealed and appreciated. Based on your and other reviewers' valuable comments and suggestions, we have now presented the manuscript with substantial revision and—hope you will agree—a significant improvement.

MAIN CONCERNS

1. There is a major disconnect between the high-level description of the results (in abstract, introduction, and discussion) and the results presented (in results section and figures). The high-level description presents complementary roles for HC and OF, with HC showing “modulation based on temporal context” and OF representing “common task structure and outcome prediction”. In contrast, the results and figures rely heavily on the “sequence-splitting” terminology (odor-, delay-, and reward-splitting neurons) and the connection between the two is hard to follow. I’m still not sure how their HC results have anything to do with “temporal context”. Even “task structure” seems to have different meanings in different parts of the manuscript. While I appreciate the effort to make the abstract and introduction more appealing to a broader audience, there needs to be a better connection with the results that are actually presented. For example, statements to the effect that HC and OF “complementarily organize associative memory” (abstract) or are responsible for “distinct components of associative memory” (discussion) and not well supported by the presented data. There are many more such examples.

Thank you for the valuable comments and for providing specific examples of our use of some terms that were not well-defined and that potentially caused clarity issues and confusion. We have now revised the manuscript based on these comments.

2. The novelty of the findings is debatable.

The rationale for the experiment can be paraphrased as “we already know a lot about the coding of HC and OF neurons, but studies comparing the two in the same behavioral task are rare”. That is technically true, but it does not make this an important study.

In the previous version of the manuscript, we phrased the comparison between the hippocampus and OFC in a way that was probably too broad. We have revised the manuscript to show that we would like to specifically compare the two brain regions on how their activities generalize between learned associations. As recognized by the reviewer, this is an important and timely problem that we think still lacks a satisfying answer, and we strongly believe the current study holds sufficient novelty to this end.

Firstly, although we could always find some hints from existing literature, the predictions can be contradictory to each other based on which hypothesis one would like to take. For instance, the hippocampal conjunctive coding suggests less generalization between learned associations, while the existence of schema cells and predictive codes in the hippocampus suggests greater generalization especially after extensive training. A direct experimental comparison is clearly needed before we can be sure about the results.

Secondly, some of the findings were beyond our expectations. For instance, the clear-cut differences between the hippocampus and OFC in splitting signal generalization suggest drastic functional differences between the brain regions, which is contrary to recent hypotheses emphasizing their functional similarities. We also found a prominent neural signature for the task schema in the OFC but not in the hippocampus, which is consistent with our own work (Zhou et al., 2021) but was not precisely predicted. Besides, without the current data, it was difficult to predict how splitting signals and their generalization would evolve during learning.

Lastly, there are obvious difficulties in direct comparisons between similar studies in the rodent hippocampus, which often use spatial tasks, and rodent OFC studies, which often use non-spatial tasks. It is not trivial to connect these studies from the two nearly distinct research fields and attempt to understand their functional differences using the same experimental settings. The current study is quite novel in successfully demonstrating the significance of doing so.

Implied here is that this study would reveal differences between HC and OF that were not already known, and it does not. Also implied is that this would be a direct comparison between HC and OF (i.e., activity recorded in both structures in the same animals), and this is not the case. To be fair, the authors do mention that the data from each region come from separate groups of animals (on line 93 and in the Supplementary Table), but such key information should come earlier and be clearer.

Thank you for the suggestion! We have now mentioned that the imaging data were collected from separate mice earlier and more explicitly in the Results (**lines 68-70 and 89-93**).

3. Clarity is a major problem with this manuscript.

I am an expert in this area, and I had to read the manuscript several times to understand it. Captions that are more narrative in nature (explaining, not just listing, what is shown) would help tremendously. A consistent and simpler terminology for the different trial types would also be important. The current terminology (S1a+, S1a-, S1b+, S1b-, S2a+, ...) is OK when there aren't too many comparisons to track (up to Fig 3) but it becomes overwhelming after that. This applies to the main text as well, since those trial types are used as examples of the contrasts of interest. I can't imagine a non-expert would be able to follow the manuscript in its current form.

Thank you for the suggestions listed here to improve the clarity of the manuscript. We have now revised the manuscript according to these suggestions.

4. The use of pseudo-ensembles (e.g., Fig 4) is questionable.

The authors recorded enough cells per animal to decode ensembles properties in each animal individually and report central tendencies at the group level. I understand that some animals had fewer cells and that some "lost" their cells at some point during testing, but there are a number of ways to aggregate data in such cases (even with basic GLMs). Pseudo-ensembles should be viewed as a last resort, as for a preparation yielding only a few cells per animals.

As mentioned by the reviewer, in some sessions, the imaged cells were fewer than in other sessions, or there were not enough in number for fair comparisons (**Supplementary Table 1**), which made it difficult to consistently perform real-ensemble analyses (meaning that neurons used for analyses came from the same imaging session and mouse), especially in the OFC, in which fewer neurons were imaged than in the CA1. Critically, to perform the coding similarity analysis—calculated as the cosine similarity between decoder hyperplanes—high decoding accuracy was desirable. To ensure high decoding accuracy, we chose to perform population-level decoding analyses using pseudo-ensembles (**lines 167-181**). However, we do recognize the importance of doing real-ensemble analyses since they help strengthen our findings. Therefore, whenever possible, we added analyses using real-ensembles. Specifically, we calculated the fractions of splitting neurons from each session and plotted results for

each subject besides the averaged fractions (**Fig. 3c-d**). For the coding similarity analysis, we replicated our analyses on individual mice and sessions (**Supplementary Fig. 13 and 16; lines 179-181 and 235-236**), and all results were highly consistent with those using pseudo-ensembles.

5. The results presented in Figures 5, 6 and 7 are very complex but seem to add little to the story. If I understand correctly, Figs 2 and 3 show similar coding properties between HC and OF. Differences between HC and OF appear in Fig 4 (panels e vs f) which shows that the black curve is shifted up for OF, reflecting more similarity between odor-splitting and delay-splitting vectors (the implication is that the coding in HC is more conjunctive). As far as I know, that's the main finding. Figures 5-7 seem to be different ways to show the same thing with more complex models -- their added value is unclear.

We are sorry for not being able to communicate the value of the data successfully, possibly due to the lack of sufficient clarity. **Fig. 5-7** are essential to answering the central question we asked, which is how the splitting signals in both brain regions generalized across learned associations. In Problem 1, there were only two sequence pairs, one for odor-splitting and another one for delay-splitting. While the coding similarity between the odor- and delay-splitting signals indeed demonstrated more generalization in the OFC, it did not show the whole picture since the two sequence pairs in Problem 1 did not contain all the task variables to be tested, especially the 'task schema' since the two sequence pairs did not share the same task schema. When all the sequence pairs were combined (i.e., mice performing Problems 1 and 2 at the same time), we could then test the neural generalization between four sequence pairs under the influences of all task variables (i.e., the past and current odor cues, task schema, and expected outcome) that reflect different aspects of the associative structure.

MINOR COMMENTS

1. To me, the discussion section read as overly speculative and broad given the findings. I also felt it should include a discussion of the limitations of the study.

We have revised the discussion and added a paragraph to discuss the limitations of the current study (**lines 332-344**).

2. In Fig 1, performance of the HC and OF groups should be shown separately.

We have plotted their behavioral performance separately in **Fig. 1** and described in lines **89-93**.

3. I understand that the cell counts per animal are shown in Supplementary Table 1, but it is customary to include a basic description of the counts (total and mean per animal or region) in the main text.

We have added such a description in the main text (**lines 89-93**).

4. Why show variability as standard deviation for 500 runs (e.g., Fig 4)? Why not confidence intervals for the curve themselves?

Based on the existing literature (e.g., Duan et al., Nat Neuro., 2021), we believe the standard deviation, in this case, is an appropriate measure for indicating the variability in the neural decoding results, although some other studies might use SEM or CI. Besides, to test the statistical significance of the decoding accuracy, we compared the mean decoding accuracy from the actual decoding with that of the 'mean decoding + 95% confidence interval' resulting from the label-shuffled decoding.

5. On that note, showing the confidence intervals for shuffled data as a line (instead of an interval) is a bit counterintuitive. Does it reflect the upper bound of the estimate (mean + CI95%)? This should be clarified.

We are sorry for the mistake, and thank you for pointing this out. Indeed, the dashed lines indicate the upper bound of the estimate (mean + CI 95%).

6. It's unclear what the blue and green traces are in Fig 4e,f.

In these plots, the green trace indicated coding similarity within the odor-splitting signals. Basically, we split the trials within trial types 1a+ and 1a- into two parts and separately trained the decoder for 1a+ vs. 1a-; we then calculated cosine similarity between the two decoder hyperplanes. The results showed a baseline similarity within the same splitting signals. Likewise, the blue trace indicated the similarity within the delay-splitting signals. We have updated these plots with clearer captions (**Fig. 4c**).

7. This part of OF appears very anterior. Is there a reason this specific segment was targeted? Does it match that of previous studies?

We targeted the lateral part of the mouse OFC since most OFC studies using rats targeted the lateral OFC, with the hope of making the current results more comparable to previous findings in rats. Therefore, the coordinates of OFC used in this study (AP 2.5 mm and ML 1.5 mm) were based on existing literature and the stereotaxic coordinates of Paxinos and Franklin (2013). Based on the atlas, the mouse OFC indeed appears more anterior than that in the rats, which does match that of previous mouse studies, for instance: (1) Namboodiri et al., Nat Neuro.,2019: vmOFC, AP 2.5 mm, (2) Banerjee et al., Nature, 2020: IOFC, AP 2.6 mm, (3) Jennings et al., Nature, 2019: OFC, AP 2.54 mm.

8. The imaging procedure resulted in considerable brain damage above the region of interest (particularly in OF; Suppl Fig 2), which I would recommend disclosing. Perhaps it would help reassure the reader to clarify that similar coding properties have been reported with electrophysiology (so the brain damage isn't a major concern)?

The diameter of GRIN lenses (1000 μm) we used for OFC imaging was considered standard in mouse cortex imaging (e.g., Namboodiri et al., Nat Neuro.,2019; Jennings et al., Nature, 2019; Banerjee et al., Nature, 2020). However, it indeed increased the chance to cause more damage than electrophysiological recordings that used electrode bundles or silicon probes, which is indeed a drawback of GRIN lens imaging. However, when compared with previous electrophysiological recordings in mice (e.g., Zhou et al., J Neuro., 2015) and rats (e.g., Stalnaker et al., Nat Commun., 2014), we found highly similar fractions of OFC neurons becoming outcome-predictive (~30%) during the delay time, suggestive of intact OFC functions. We also added discussions about the limitations of using GRIN-lens imaging for neural correlative studies; for instance, future studies using electrophysiology would be helpful to confirm the current findings further (**lines 332-338**).

9. A couple of papers by Ramus are particularly relevant to this paper:

Ramus & Eichenbaum (2000) Neural correlates of olfactory recognition memory in the rat orbitofrontal cortex. J Neurosci: 20, 8199-208. <https://doi.org/10.1523/jneurosci.20-21-08199.2000>

Ginther MR, Walsh DF, Ramus SJ (2011) Hippocampal neurons encode different episodes in an overlapping sequence of odors task. *J Neurosci* : 31(7), 2706-11. doi: 10.1523/JNEUROSCI.3413-10.2011.

Thank you for the reminder of these papers! We have added them to the revised manuscript.

References

Friedrich, J., Zhou, P., & Paninski, L. (2017). Fast online deconvolution of calcium imaging data. *PLOS Comput Biol*, 13(3), e1005423.

Kinsky, N. R., Mau, W., Sullivan, D. W., Levy, S. J., Ruesch, E. A., & Hasselmo, M. E. (2020). Trajectory-modulated hippocampal neurons persist throughout memory-guided navigation. *Nat Commun*, 11(1), 2443.

Chen, T. W., Wardill, T. J., Sun, Y., Pulver, S. R., Renninger, S. L., Baohan, A., ... & Kim, D. S. (2013). Ultrasensitive fluorescent proteins for imaging neuronal activity. *Nature*, 499(7458), 295-300.

Duan, C. A., Pagan, M., Piet, A. T., Kopec, C. D., Akrami, A., Riordan, A. J., ... & Brody, C. D. (2021). Collicular circuits for flexible sensorimotor routing. *Nat Neuro*, 24(8), 1110-1120.

Namboodiri, V. M. K., Otis, J. M., van Heeswijk, K., Voets, E. S., Alghorazi, R. A., Rodriguez-Romaguera, J., ... & Stuber, G. D. (2019). Single-cell activity tracking reveals that orbitofrontal neurons acquire and maintain a long-term memory to guide behavioral adaptation. *Nat Neuro*, 22(7), 1110-1121.

Jennings, J. H., Kim, C. K., Marshel, J. H., Raffiee, M., Ye, L., Quirin, S., ... & Deisseroth, K. (2019). Interacting neural ensembles in orbitofrontal cortex for social and feeding behaviour. *Nature*, 565(7741), 645-649.

Banerjee, A., Parente, G., Teutsch, J., Lewis, C., Voigt, F. F., & Helmchen, F. (2020). Value-guided remapping of sensory cortex by lateral orbitofrontal cortex. *Nature*, 585(7824), 245-250.

Zhou, J., Jia, C., Feng, Q., Bao, J., & Luo, M. (2015). Prospective coding of dorsal raphe reward signals by the orbitofrontal cortex. *J Neurosci*, 35(6), 2717-2730.

Stalnaker, T. A., Cooch, N. K., McDannald, M. A., Liu, T. L., Wied, H., & Schoenbaum, G. (2014). Orbitofrontal neurons infer the value and identity of predicted outcomes. *Nat Commun*, 5(1), 3926.

Zhou, J., Jia, C., Montesinos-Cartagena, M., Gardner, M. P., Zong, W., & Schoenbaum, G. (2021). Evolving schema representations in orbitofrontal ensembles during learning. *Nature*, 590(7847), 606-611.

Reviewer #1 (Remarks to the Author):

It is difficult to find the precise text changes within the manuscript which have been requested to improve clarity of the results. I would have appreciated a labelling of altered relevant text chapters or mentioning of examples within the response letter.

Reviewer #1 (Remarks on code availability):

NA

Reviewer #2 (Remarks to the Author):

The authors did a good job of addressing my comments from the first round of review. In doing so, the manuscript has been improved. I have no further comments.

Reviewer #2 (Remarks on code availability):

n/a

Reviewer #3 (Remarks to the Author):

I thank the authors for improving the clarity of the introduction and the results. However, I still have major concerns about the novelty of the findings and the interpretation of the results.

1. The novelty of the findings is still debatable. The main result of the paper is that hippocampal ensembles are more conjunctive than those in OFC. In other words, hippocampal activity tends to be more sequence-specific (e.g., odor 3 when preceded by odor 1 but not by odor 2), trial-type-specific (odor-splitting vs delay-splitting), and problem-specific (1 vs 2) than those in OFC. This may not be the most exciting way to present the results, but it is the most parsimonious and accurate way. Yes, it is probably the first time that a hippocampus group and an OFC group were tested on the same task, but that doesn't mean the findings are novel (see below).

2. The interpretation of the results is still concerning. While I understand the need to "sell" the results for a top journal, interpreting the hippocampal results in terms of a temporal context effect is incorrect (see 3 below) and the OFC results in terms of schemas is a massive overreach given the data (see 4). This framing of the results, and the speculative discussion extensively expanding on those topics, gives the reader the impression that these important issues have been addressed here and they really haven't.

3. The hippocampal results are consistent with the large literature on conjunctive and context-specific coding in the hippocampus, such as the "splitter" cells for spatial and non-spatial information (now added to the discussion). The temporal context interpretation does not fit well here. If there is a connection there (the authors point to a review by Duvelle et al 2023), it needs to be made explicit. The task can be viewed as a series of paired-associates (conditional associations; e.g., AB \diamond reward, CB \diamond No reward). Yes, some of the paired associates are separated by time, but that doesn't mean it taps into temporal context mechanisms.

4. The logic in the interpretation of the OFC results is flawed. The authors show that OFC coding is less conjunctive (or context-specific) than hippocampus coding. I cannot think of a structure in the brain that has shown more conjunctive (or context-specific) coding than the hippocampus; therefore, any other structure compared to it will show less conjunctive coding. This is providing the basic context for interpreting a result and is not done appropriately here. Also, evoking schemas is an overreach for the OFC results presented. For example, if a group of neurons tends to respond to an outcome (like a reward or expectation of a reward) across task conditions or contexts, it doesn't necessarily mean that they abstracted the complex shared structure across such conditions (which a schema would imply). It could simply mean that they tend to respond to an outcome (i.e., they are not sensitive to condition information). The latter is the more

parsimonious and, presumably, more accurate explanation. The authors seem to throw many analyses at this issue (Fig 5-7) but, if one looks carefully, there is little here to disprove that simpler explanation.

4. Comparative statements cannot be made by comparing p values from two separate tests (e.g., Results, Line 123-126). The analysis must include data from both and test for interactions. See Nieuwenhuis et al (2011) Nature neuro: 14, 1105-1107.

5. I still don't love the pseudo-ensembles approach (and interpretations based on downsampling) but I appreciate that the authors added analyses from real ensembles.

Minor comments

- Results, Line 113. When reporting fractions, here and elsewhere, the denominator should be specified. Is it relative to all cells imaged? All cells active in the task or trials?...

- Fig 1 caption. "(0-7s from the onset of the first odor)" is a bit confusing as it could imply before the trial. I would suggest specifying that 0 is odor onset.

- Fig 1 panels e,f. Is the example data shown from a trial epoch? It should be.

- Fig 4 caption (and others). When referring to the dotted line, I would suggest directly saying it is the upper bound of the confidence interval otherwise it may be confusing to the reader. e.g. "The dotted line refers to the upper bound of the 95% confidence interval (mean + 95%CI)".

REVIEWER COMMENTS

Reviewer #1 (Remarks to the Author):

It is difficult to find the precise text changes within the manuscript which have been requested to improve clarity of the results. I would have appreciated a labelling of altered relevant text chapters or mentioning of examples within the response letter.

Thank you for your time reviewing our paper. We have labeled relevant text changes in the revised manuscript and explained the changes below.

Reviewer #2 (Remarks to the Author):

The authors did a good job of addressing my comments from the first round of review. In doing so, the manuscript has been improved. I have no further comments.

Thank you! We appreciate the time in evaluating our work and providing invaluable comments and suggestions that have improved the manuscript.

Reviewer #3 (Remarks to the Author):

I thank the authors for improving the clarity of the introduction and the results. However, I still have major concerns about the novelty of the findings and the interpretation of the results.

Thank you for acknowledging the improved clarity of the manuscript. We greatly value your expertise and critical evaluation of our work and appreciate your insights that helped us further improve the manuscript. While we firmly believe the current study's novelty and value to a broad audience who are interested in understanding the roles of two important brain regions, after carefully reading your comments, we feel there might be places in the manuscript that were not clear enough, so we have further revised the manuscript to resolve these concerns and provide detailed responses below.

1. The novelty of the findings is still debatable. The main result of the paper is that hippocampal ensembles are more conjunctive than those in OFC. In other words, hippocampal activity tends to be more sequence-specific (e.g., odor 3 when preceded by odor 1 but not by odor 2), trial-type-specific (odor-splitting vs delay-splitting), and problem-specific (1 vs 2) than those in OFC. This may not be the most exciting way to present the results, but it is the most parsimonious and accurate way. Yes, it is probably the first time that a hippocampus group and an OFC group were tested on the same task, but that doesn't mean the findings are novel (see below).

From the above, it seems that we did a poor job describing our results regarding the differences between hippocampus and OFC in the sequence-specific activity. The reviewer's summary states that the hippocampal activity was more sequence-specific; however, our results actually show the opposite. That is, the OFC activity was actually more sequence-specific, i.e., more neurons showed sequence-specific activities to the common odor (e.g., Odor 3) or the common delay. For instance, on Day 12,

~35% of neurons in OFC exhibited differential activities to Odor 3 preceded by different leading odors, but ~20% in CA1 (**Figure 3c-d, lines 120-123**). The meanings of ‘conjunctive coding’ might vary based on context. Suppose the conjunctive coding was defined as the sequence-specific activity per se—meaning differential neural responses to the same current sensory input but with different preceding cues. In that case, the comparison suggests that the OFC has even stronger conjunctive coding than the hippocampus.

However, this is not the kind of ‘conjunctive coding’ we were interested in. In this study, we specifically focused on a related but, we believe, far more important and unexplored question—how sequence-specific activity (i.e., sequence-splitting signals) generalized across sequence pairs under the influence of four task variables, i.e., the past and current cues, sequence structure (previously called ‘task schema’, meaning how the ‘cue-outcome’ was arranged—basically with or without a second odor cue), and expected outcome. In other words, we aimed to test whether the splitting signal associated with one sequence pair also split another sequence pair and how such generalization was affected by these task variables. Therefore, in our opinion, a more accurate summary of our finding is that the generalization of sequence-splitting signals in the hippocampus depended more on proximal or current sensory cues and those in OFC more on the sequence structure and expected outcome.

Here is a summary of the most important messages in the manuscript, which we hope our revisions now convey more clearly.

(1) A desirable and timely study in the field

Our major findings came from examining the splitting signal generalization, which we believe is a research interest not exclusively our own but shared with other people in the field as well. For instance, during our collection of current data, Duvelle et al. published an excellent review paper in eLife in 2023. The authors propose experiments similar to ours but in the spatial domain (Duvelle et al., eLife, 2023; Figure 3, many T-maze task) that aim to investigate the roles of hippocampal splitter signals in signaling temporal context and hidden-state inference by examining splitter signal generalization. The rationale is that the hidden-state inference model predicts more generalized splitter signals than the temporal context model, which is consistent with a recent finding (Sun et al., Nat Neurosci, 2020) showing that 38% of the ‘lap-counter’ cells (conjunctive coding of position and lap count) generalize from one maze to another, however inconsistent with other studies (Hallock and Griffin, Hippocampus, 2013; Bahar and Shapiro, J Neurosci, 2012) showing few consistent splitters across slightly different versions of the same maze. Thus, these mixed results raise a critical but unexplored question: how is the generalization of the splitting signals affected under different task conditions? By using carefully designed task and data analyses, we were able to quantify and compare the influences of four task variables on splitting signal generalization in the hippocampus and OFC. Although more questions remain, we believe our current report is desirable and would make a timely contribution to the field.

(2) A new type of conjunctive coding in the hippocampus

Our findings are consistent with known conjunctive coding in the hippocampus. However, they also extend our knowledge by uncovering a new type of conjunctive coding—the conjunction between the learning-related splitting signals, proximal and distal sensory cues, and sequence structure (previously

called ‘task schema’) in a nonspatial task, which some researchers might expect to see but currently not available in the literature.

More specifically, past observations on hippocampal conjunctive coding mainly focus on the formation of a unique neural code for a combination of observable physical variables, i.e., a particular set of bottom-up sensory stimuli like the sensory cue, reward, and time within a spatial (e.g., Komorowski et al., *J Neurosci*, 2009) and non-spatial (e.g., Yadav et al., *Nature*, 2022) environment. To the best of our knowledge, the conjunctive coding of the type we report here has not been shown in the hippocampus. One study that is most relevant to our finding was published by Nieh et al., in *Nature* in 2021, in which they find a hippocampal conjunctive coding of a cognitive variable (i.e., accumulated evidence) and physical space. Still, our findings are novel because 1) other than the passively accumulated evidence, the splitting signals in our study emerged due to learning, and 2) our task was nonspatial thus likely revealed more general hippocampal functions.

(3) Necessity to directly compare between hippocampus and OFC

As the reviewer notes, there are almost no direct comparisons between the hippocampus and OFC—the same task, same lab, same learning conditions, same analyses—and this is really critical because 1) increasingly, their functions seem to be quite convergent within or among the groups of brain regions people study and 2) one simply cannot draw firm conclusions from differences like those we report here from speculating about very different studies.

(4) More parsimonious accounts for the seemingly predictive signals

Differential neural activity to CS+ vs. CS- after cue-outcome learning is widely observed across the brain, and this activity is commonly thought of as an outcome-predictive signal. Our results showed that while such neural activity indeed resulted from associative learning (**Figure 3**), its meaning can be different across brain regions (**Figure 7**). We were able to provide critical insights into this because the novel task design and data analyses allowed us to decompose the recorded neural activity and identify particular contributions of different task variables to this neural signal.

(5) Neural correlations consistent with schema representations

Both brain regions have been shown to encode schemas in various settings (e.g., Tse et al., *Science*, 2007; McKenzie et al., *Neuron*, 2014; Baraduc et al., *Science*, 2019; Zhou et al., *Nature*, 2021). More experiments are needed to compare them in the same task conditions. Although lacking causal evidence, we consider that the neural correlation of sequence structure is related to this because, consistent with a schema representation, it predicts stronger generalization between sequence pairs sharing the same sequence structure, which the ‘expected outcome’ does not because all sequence pairs share the same expected outcome.

(6) Complementary contributions of the hippocampus and OFC

The study of interactions between the hippocampus and the prefrontal regions is a hot topic of research (e.g., Preston and Eichenbaum, *Curr Biol*, 2014; Yadav et al., *Nature*, 2022). Complementary or concurrent processes between the hippocampus and OFC have been proposed and reported in other behavioral settings (Zhou et al., *Curr Biol*, 2019; Mızrak et al., *Cell Rep*, 2023) but not in tasks like ours

to reveal different aspects of learned associations. Thus, our finding of complementary contributions of the hippocampus and OFC provides novel insights and promotes future testing of the interdependence of the two brain regions in establishing and organizing the most basic form of learned associations.

2. The interpretation of the results is still concerning. While I understand the need to “sell” the results for a top journal, interpreting the hippocampal results in terms of a temporal context effect is incorrect (see 3 below) and the OFC results in terms of schemas is a massive overreach given the data (see 4). This framing of the results, and the speculative discussion extensively expanding on those topics, gives the reader the impression that these important issues have been addressed here and they really haven't.

We appreciate the reviewer's comments and concerns about the interpretations of the results. Our responses to the interpretation of the hippocampal and OFC results in terms of temporal context and task schema, respectively, are provided below (see these responses under 3 and 4).

3. The hippocampal results are consistent with the large literature on conjunctive and context-specific coding in the hippocampus, such as the “splitter” cells for spatial and non-spatial information (now added to the discussion). The temporal context interpretation does not fit well here. If there is a connection there (the authors point to a review by Duvelle et al 2023), it needs to be made explicit. The task can be viewed as a series of paired-associates (conditional associations; e.g., AB \diamond reward, CB \diamond No reward). Yes, some of the paired associates are separated by time, but that doesn't mean it taps into temporal context mechanisms.

(1) The ‘temporal context’ account

After considering the reviewer's comments, we agree that the support for ‘temporal context’ was insufficient. In the previous version of the revised manuscript, we toned down the use of ‘temporal context’. We were not to assert that the hippocampus represents ‘temporal context’ but rather that our findings align with this view.

More specifically, as the reviewer notes, the task can be viewed as paired associates (e.g., AB \rightarrow reward, CB \rightarrow no reward). However, the first (past) and second (current) odor cues were separated in time, and this is essential to the temporal context model (TCM), which proposes an integration of the current stimuli and temporally delaying past stimuli. The TCM was relevant here because we observed a stronger influence from the current than the past odor cues on the splitting signal generalization in the hippocampus (higher estimated β than α in CA1, **Figure 7, lines 248-251**). Such temporally decaying influence from sensory cues is a characteristic of TCM (Howard and Kahana, J Math Psychol, 2002). However, since we did not formalize our model to recapitulate all aspects of TCM (e.g., contextual drift rate), our current results were indeed insufficient to support that the hippocampus represents ‘temporal context’. Instead, we concluded that our findings were in line with this view.

Nevertheless, and more importantly, the significance of our hippocampal results does not hinge on the ‘temporal context’ account. To further clarify this, we substituted ‘temporal context’ with ‘recent sensory experience’ or ‘proximal sensory cues’ where necessary to interpret our results more objectively—which is that the hippocampal splitting signals were more dependent on recent sensory experiences, and the OFC splitting signals tended to be more generalizable based on sequence pair similarities despite of

distinct sensory inputs. We have also scrutinized the manuscript to identify places where the writing was too speculative and made changes accordingly.

(2) The conjunctive coding

The reviewer mentions again that our results are consistent with the current literature that hippocampal neurons exhibit conjunctive coding, particularly in spatial tasks. This widely accepted understanding has actually contributed to shaping our current hypothesis, which is that the hippocampal conjunctive coding can be further extended to seemingly 'predictive signals' in a nonspatial task.

Another crucial point we hope to include is that, beyond the established hippocampal conjunctive coding, recent studies increasingly indicate the role of the hippocampus in prediction, generalization, abstraction, and schema representations typically associated with prefrontal regions like the OFC; on the other hand, the OFC has also been suggested to be necessary for encoding cognitive maps and for hidden-state inference previously linked to the function of the hippocampus. This convergence in how people describe the functions of these two brain regions is intriguing and attractive yet quite puzzling. Again, without direct comparisons in identical conditions, definitively answering questions about the functional differences between these brain regions would be difficult. Therefore, no matter whether the results are expected or not, our work is novel and significant as it represents a starter of such needed research, and we believe this work would be appreciated by a broad audience who are interested in the functional differences and connections between the hippocampus and OFC.

4. The logic in the interpretation of the OFC results is flawed. The authors show that OFC coding is less conjunctive (or context-specific) than hippocampus coding. I cannot think of a structure in the brain that has shown more conjunctive (or context-specific) coding than the hippocampus; therefore, any other structure compared to it will show less conjunctive coding. This is providing the basic context for interpreting a result and is not done appropriately here. Also, evoking schemas is an overreach for the OFC results presented. For example, if a group of neurons tends to respond to an outcome (like a reward or expectation of a reward) across task conditions or contexts, it doesn't necessarily mean that they abstracted the complex shared structure across such conditions (which a schema would imply). It could simply mean that they tend to respond to an outcome (i.e., they are not sensitive to condition information). The latter is the more parsimonious and, presumably, more accurate explanation. The authors seem to throw many analyses at this issue (Fig 5-7) but, if one looks carefully, there is little here to disprove that simpler explanation.

(1) The conjunctive coding

We agree. We cannot think of other areas either that have been shown to be more conjunctive than the hippocampus. And yet here we show that OFC, by some definitions, may be one such area. As already described in our response to reviewer's comment #1, for instance, if sequence-specific neural activity (i.e., sequence-splitting signal) per se was seen as conjunctive coding, there were even more sequence-specific neurons, thus more conjunctive coding, in the OFC than in the hippocampus. However, supposing the conjunctive coding is about how task variables like past sensory cues affect splitting signal generalization between sequence pairs, our results showed stronger conjunctive coding

in the hippocampus than in the OFC. We hope the reviewer will appreciate the novelty of our results in this light. Please see the detailed discussions on this issue in the responses to the comments above.

(2) The definition of 'task schema'

From our reading of the comment, we speculate we were unclear in our definition of task schema. The use of 'task schema' was intended to describe a particular 'cue-outcome' sequence arrangement (i.e., 'sequence structure'), basically, whether there was a second odor cue in between the first odor cue and the outcome. Specifically, the sequence pair 1a+ vs. 1a- and another sequence pair 2a+ vs. 2a- shared the same sequence structure (i.e., sequence pair type a) because of the existence of the second odor cues, while the sequence pair 1b+ vs. 1b- and another sequence pair 2b+ vs. 2b- shared the same sequence structure (i.e., sequence pair type b) because of the existence of the long delay. Thanks to the reviewer's comment, we realized that our use of 'task schema' might be misleading. Therefore, we replaced the term with 'sequence structure' to refer to this critical task variable more objectively. We agree that such a neural correlation for the 'sequence structure' does not equal schema representation, but it shows consistency in terms of how they would affect splitting signal generalization.

In addition, we did not consider the context-independent response to an outcome as evidence that a neuron encodes 'task schema' or 'sequence structure'. Our results showed that the OFC splitting signals were more generalized between sequence pairs that shared the same sequence structure than between sequence pairs that differed in sequence structures. Such a pattern of neural activity related to sequence structure cannot be explained by 'expected outcome' or other task variables. For instance, since all of the four sequence pairs shared the same outcomes, the 'expected outcome' account would predict nonselective generalization across all splitting signals. In other words, the two task variables—'sequence structure' and 'expected outcome'—contributed differently to splitting signal generalization.

4. Comparative statements cannot be made by comparing p values from two separate tests (e.g., Results, Line 123-126). The analysis must include data from both and test for interactions. See Nieuwenhuis et al (2011) Nature neuro: 14, 1105-1107.

Thank you for raising this important statistical issue. We have updated the manuscript with new analyses that tested interactions (**lines 120-123, 125-127, 177-179, 181-183**).

5. I still don't love the pseudo-ensembles approach (and interpretations based on downsampling) but I appreciate that the authors added analyses from real ensembles.

Thank you for your understanding. Due to the low numbers of the subjects (n = 7 mice, hippocampus; n = 4 mice, OFC), we think the pseudo-ensembles approach is more rigorous in this case.

Minor comments

- Results, Line 113. When reporting fractions, here and elsewhere, the denominator should be specified. Is it relative to all cells imaged? All cells active in the task or trials?...

Thank you for the questions. We have now specified the denominator in the figure caption (**Figure 3, lines 742-743**). Specifically for the Results in Line 113, the denominator was all neurons imaged on a given day, same as the fractions reported elsewhere unless otherwise specified.

- Fig 1 caption. “(0-7s from the onset of the first odor)” is a bit confusing as it could imply before the trial. I would suggest specifying that 0 is odor onset.

We now specified that 0 is the time of odor onset in the figure caption (**Figure 1, lines 729-730**).

- Fig 1 panels e,f. Is the example data shown from a trial epoch? It should be.

These example data were not aligned to one trial epoch but spanning consecutive 500 seconds within a session (**Figure 1 caption and lines 730-734**). We used such a demonstration, including the calcium-indicated fluorescent signals and deconvoluted spike rates, to provide a glimpse of the quality of our imaging data and an intuitive impression of the neural data to be analyzed. We did not show calcium transients aligned to a single trial epoch because they would be too sparse for this type of visual demonstration. However, examples of trial epoch-aligned neural activities are presented in **Figure 2**.

- Fig 4 caption (and others). When referring to the dotted line, I would suggest directly saying it is the upper bound of the confidence interval otherwise it may be confusing to the reader. e.g. “The dotted line refers to the upper bound of the 95% confidence interval (mean + 95%CI)”.

Done. Changes are in figure captions and main text (**Figures 4 and 6, lines 758-759, 762, 779-780**).

Thanks again for the helpful comments from the reviewers. After additional revisions, we believe the current manuscript communicates our findings more accurately.

References

Duvelle, É., Grieves, R. M. & van der Meer, M. A. Temporal context and latent state inference in the hippocampal splitter signal. *eLife* 12, e82357 (2023).

Sun, C., Yang, W., Martin, J. & Tonegawa, S. Hippocampal neurons represent events as transferable units of experience. *Nat. Neurosci.* 23, 651–663 (2020).

Hallock, H. L. & Griffin, A. L. Dynamic coding of dorsal hippocampal neurons between tasks that differ in structure and memory demand. *Hippocampus* 23, 169–186 (2013).

Bahar, A. S. & Shapiro, M. L. Remembering to Learn: Independent Place and Journey Coding Mechanisms Contribute to Memory Transfer. *J. Neurosci.* 32, 2191–2203 (2012).

Komorowski, R. W., Manns, J. R. & Eichenbaum, H. Robust conjunctive item–place coding by hippocampal neurons parallels learning what happens where. *J. Neurosci.* 29, 9918–9929 (2009).

Yadav, N. et al. Prefrontal feature representations drive memory recall. *Nature* (2022)
doi:10.1038/s41586-022-04936-2.

Nieh, E. H. et al. Geometry of abstract learned knowledge in the hippocampus. *Nature* 595, 80–84 (2021).

Tse, D., Langston, R. F., Kakeyama, M., Bethus, I., Spooner, P. A., Wood, E. R., ... & Morris, R. G. (2007). Schemas and memory consolidation. *Science*, 316(5821), 76-82.

McKenzie, S. et al. Hippocampal representation of related and opposing memories develop within distinct, hierarchically organized neural schemas. *Neuron* 83, 202–215 (2014).

Baraduc, P., Duhamel, J. R., & Wirth, S. (2019). Schema cells in the macaque hippocampus. *Science*, 363(6427), 635-639.

Zhou, J. et al. Evolving schema representations in orbitofrontal ensembles during learning. *Nature* 590, 606–611 (2021).

Preston, A. R. & Eichenbaum, H. Interplay of Hippocampus and Prefrontal Cortex in Memory. *Curr. Biol.* 23, R764–R773 (2013).

Zhou, J. et al. Complementary task structure representations in hippocampus and orbitofrontal cortex during an odor sequence task. *Curr. Biol.* 29, 3402–3409 (2019).

Mızrak, E., Bouffard, N. R., Libby, L. A., Boorman, E. D. & Ranganath, C. The hippocampus and orbitofrontal cortex jointly represent task structure during memory-guided decision making. *Cell Rep.* 37, 110065 (2021).

Howard, M. W. & Kahana, M. J. A distributed representation of temporal context. *J. Math. Psychol.* 46, 269–299 (2002).